# Near-Optimal Regret-Queue Length Tradeoff in Online Learning for Two-Sided Markets

**Zixian Yang**
Department of Electrical Engineering and Computer Science
University of Michigan
Ann Arbor, MI 48109
`zixian@umich.edu`

**Sushil Mahavir Varma**
Department of Industrial and Operations Engineering
University of Michigan
Ann Arbor, MI 48109
`sushilv@umich.edu`

**Lei Ying**
Department of Electrical Engineering and Computer Science
University of Michigan
Ann Arbor, MI 48109
`leiying@umich.edu`

## Abstract

We study a two-sided market, wherein, price-sensitive heterogeneous customers and servers arrive and join their respective queues. A compatible customer-server pair can then be matched by the platform, at which point, they leave the system. Our objective is to design pricing and matching algorithms that maximize the platform's profit, while maintaining reasonable queue lengths. As the demand and supply curves governing the price-dependent arrival rates may not be known in practice, we design a novel online-learning-based pricing policy and establish its near-optimality. In particular, we prove a tradeoff among three performance metrics: $\tilde{O}(T^{1-\gamma})$ regret, $\tilde{O}(T^{\gamma/2})$ average queue length, and $\tilde{O}(T^{\gamma})$ maximum queue length for $\gamma \in (0, 1/6]$, significantly improving over existing results [1]. Moreover, barring the permissible range of $\gamma$, we show that this trade-off between regret and average queue length is optimal up to logarithmic factors under a class of policies, matching the optimal one as in [2] which assumes the demand and supply curves to be known. Our proposed policy has two noteworthy features: a dynamic component that optimizes the tradeoff between low regret and small queue lengths; and a probabilistic component that resolves the tension between obtaining useful samples for fast learning and maintaining small queue lengths.

## 1 Introduction

We study a two-sided market, wherein, heterogeneous customers and servers arrive into the system and join their respective queues. Any compatible customer-server pair can then be matched, at which point they leave the system instantaneously. More formally, the system is described by a bipartite graph, where the vertices are customer and server queues while the edges represent compatibility between them.

39th Conference on Neural Information Processing Systems (NeurIPS 2025).

Such a canonical two-sided/matching queueing model is useful in studying matching in emerging applications like online marketplaces [2]. For example, matching customers and drivers in ride-hailing systems, customers and couriers in meal-delivery platforms, tasks and workers on crowdsourcing platforms, etc. Furthermore, matching queues are a versatile tool for modeling diverse applications beyond online marketplaces. In particular, different variants of matching queues are prevalent in modeling applications like assemble-to-order systems [3], payment channel networks [4], and quantum switches [5].

Motivated by these applications, we study matching in a two-sided queueing model with an additional lever of modulating the arrival rates via pricing. For example, a higher price offered to the customer or a lower price offered to the server reduces their arrival rate. Our objective is to devise a joint dynamic pricing and matching policy to maximize the profit for the system operator and minimize the delay for customers and servers. Specifically, we focus on characterizing the Pareto frontier of regret and average queue length, where regret is defined as the loss in profit compared to the so-called fluid benchmark (see (5) for a precise definition).

This model was first introduced and studied in [2]. They proposed a dynamic pricing and matching policy and established a $\Theta(\eta^{1-\gamma})$ regret and $\Theta(\eta^{\gamma/2})$ queue length for $\gamma \in [0, 1]$ in the steady state, where $\eta$ is the system size. They also showed that such a trade-off between regret and queue length is the best possible for a large class of policies. However, the proposed pricing policy of [2] is based on the optimal solution of the so-called fluid problem which relies on knowing the demand and supply curves. These curves are typically estimated using historical data. Thus, any estimation error leads to an error in the optimal fluid solution, resulting in a $\Theta(\eta)$ regret in the steady state. So, it is essential to simultaneously estimate the demand and supply curves and optimize the pricing policy.

Building on [2], the paper [1] proposed a learning-based pricing policy that iteratively updates the price based on a zeroth-order stochastic projected gradient ascent on the fluid optimization problem. They establish a $\tilde{\Theta}(T^{1-\gamma})$ regret and $\Theta(T^{\gamma})$ average queue length bounds for $\gamma \in [0, 1/6]$, where $T$ is the time horizon (plays a similar role as $\eta$ in [2]). So, while this pricing scheme is parameter-agnostic, it is not Pareto optimal as it does not achieve $1 - \gamma$ versus $\gamma/2$ tradeoff. One can understand this sub-optimality by observing that their pricing scheme mimics a static policy as $T \to \infty$ as opposed to a two-price policy as in [2].

To this end, we develop a pricing policy that achieves the best of both worlds: a parameter-agnostic pricing scheme that mimics a two-price policy. We establish an improved $\tilde{\Theta}(T^{1-\gamma})$ regret and $\tilde{\Theta}(T^{\gamma/2})$ average queue length for $\gamma \in [0, 1/6]$. We show that this trade-off, that is, $1 - \gamma$ versus $\gamma/2$ as in [2], is optimal up to logarithmic factors under a class of policies. The only discrepancy is in the permissible value of $\gamma$. In particular, we incur an additional learning error compared to [2] which prevents us from improving the regret beyond $\tilde{\Theta}(T^{5/6})$. Nonetheless, we significantly improve over [1] and achieve a near-optimal trade-off between regret and average queue length. Additionally, our policy also allows us to ensure $\Theta(T^{\gamma})$ maximum queue length.

Our pricing scheme is a novel dynamic probabilistic policy that optimizes the tension between obtaining useful samples for fast learning and maintaining small queue lengths while incurring low regret. At the time $t \in [T]$, let $p(t)$ be the price prescribed by the learning scheme for a fixed customer type with queue length $Q(t) \in \mathbb{Z}_+$. Then, we are inclined to set a price $p(t)$ to obtain useful samples for learning, but that may increase the queue length. Our policy resolves this difficulty as follows. If $Q(t) = 0$, set the price to be $p(t)$ to prioritize learning when the queue length is empty. Similarly, if $Q(t) \geqslant q^{\text{th}}$, set the price to be the maximum price to prioritize high negative drift when the queue length is large. For the in-between case of $0 < Q(t) < q^{\text{th}}$, we set the price to be $p(t)$ with probability $1/2$ and $p(t) + \alpha$ otherwise. Such a probabilistic scheme ensures that half of the collected samples is useful in learning. In addition, $\alpha \in \mathbb{R}$ is suitably picked to optimize the trade-off between maintaining low queue lengths and incurring a small regret. In other words, we converge to an optimal two-price policy as $T \to \infty$, as opposed to a static policy in [1] which is known to be sub-optimal.

## 1.1 Related Work

Contrary to the classical literature on online learning [6, 7, 8, 9, 10], learning in queueing systems poses an additional challenge of strong correlations over time induced by maintaining a queue, in

addition to its countable state space invoking the curse of dimensionality. A recent surge of literature [11, 1, 12, 13, 14] has emerged to better understand online learning in queueing.

One line of work views the queueing system as a Markov decision process and uses reinforcement learning methods in an attempt to learn a global optimal policy [15, 16]. These methods are broadly applicable to many types of queueing systems. Contrary to this, our focused approach allows us to exploit the structural result that a two-price policy is near optimal [2] to restrict our policy space, resulting in a more efficient and practical learning scheme.

In line with our approach, another line of work exploits the structure of either the model [17], the stationary distribution [18], or the (near) optimal policy [19, 13]. Closest to ours is the paper [19] that learns to price for a queueing system with a single server and a single queue, where arrivals follow a Poisson process and service times are independent and identically distributed with a general distribution. However, they assume uniform stability of the queueing system, circumventing one of the main challenges we tackle in this work. This assumption allows them to learn a static pricing policy and show its optimality [20]. On the other hand, we focus on learning a dynamic pricing policy and demonstrate its benefits, which we believe is novel in the literature on learning in queues.

## 2 Problem Formulation

We are using the model developed by [1]. We present the model in this section for completeness. We study a discrete-time system that contains two sets (sides) of queues: one for customers and the other for servers. Each side includes multiple queues to represent various types of customers or servers. Customers can be seen as demand, and servers as supply. This system can be modeled using a bipartite graph $G(\mathcal{I} \cup \mathcal{J}, \mathcal{E})$, where $\mathcal{I} = \{1, 2, \ldots, I\}$ denotes customer types and $\mathcal{J} = \{1, 2, \ldots, J\}$ denotes server types, with $|\mathcal{I}| = I$ and $|\mathcal{J}| = J$. The set $\mathcal{E}$ includes all compatible links, that is, a type $i$ customer can be served by a type $j$ server if and only if $(i, j) \in \mathcal{E}$. Figure 1 illustrates this with three customer types and two server types ($I = 3, J = 2$).

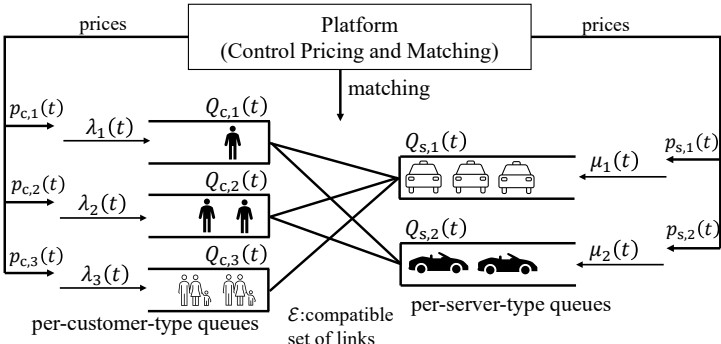

Figure 1: The model, an example with 3 types of customers and 2 types of servers.

At the beginning of each time slot $t$, the platform observes $Q_{c,i}(t)$ and $Q_{s,j}(t)$ for all $i \in \mathcal{I}$ and $j \in \mathcal{J}$, where $Q_{c,i}(t)$ and $Q_{s,j}(t)$ represent the queue lengths of type $i$ customers and type $j$ servers, respectively. Assume that all queues are initially empty at $t = 1$. Then the platform determines the price for each queue, i.e., for each type of customer and server. Let $p_{c,i}(t)$ represent the price for a type $i$ customer and $p_{s,j}(t)$ the price for a type $j$ server in time slot $t$. The price $p_{c,i}(t)$ means that a type $i$ customer will be charged $p_{c,i}(t)$ units of money by the platform when they join the queue. The price $p_{s,j}(t)$ means that a type $j$ server will receive $p_{s,j}(t)$ units of money from the platform when they join the queue. These prices will influence the arrivals of the customers and servers. In time slot $t$, let $A_{c,i}(t)$ represent the number of type $i$ customer arrivals, and $A_{s,j}(t)$ the number of type $j$ server arrivals. Assume $A_{c,i}(t)$ and $A_{s,j}(t)$ are independently distributed according to Bernoulli distributions. Define $\lambda_i(t) := \mathbb{E}[A_{c,i}(t)]$ and $\mu_j(t) := \mathbb{E}[A_{s,j}(t)]$. We use a demand function and a supply function to model the relation between the price and the arrival rate for each customer and each server. For a type $i$ customer, the demand function, $F_i : [0, 1] \to [p_{c,i,\min}, p_{c,i,\max}]$, models the relation between the arrival rate $\lambda_i(t) \in [0, 1]$ and the price $p_{c,i}(t) \in [p_{c,i,\min}, p_{c,i,\max}]$, with $p_{c,i,\min}$ and $p_{c,i,\max}$ as the minimum and maximum prices, respectively. Similarly, the supply function for a type $j$ server, $G_j : [0, 1] \to [p_{s,j,\min}, p_{s,j,\max}]$, models the relation between the arrival rate

$\mu_j(t) \in [0, 1]$ and the price $p_{s,j}(t) \in [p_{s,j,\min}, p_{s,j,\max}]$, where $p_{s,j,\min}$ and $p_{s,j,\max}$ are the minimum and maximum prices for a type $j$ server. We make the following assumption on $F_i$ and $G_j$ as in [1]:

**Assumption 1.** *For any customer type $i$, $F_i$ is strictly decreasing, bijective, and $L_{F_i}$-Lipschitz. Its inverse function $F_i^{-1}$ is $L_{F_i^{-1}}$-Lipschitz. For any server type $j$, $G_j$ is strictly increasing, bijective, and $L_{G_j}$-Lipschitz. Its inverse function $G_j^{-1}$ is $L_{G_j^{-1}}$-Lipschitz.*

Thus, the price $p_{c,i}(t)$ determines the arrival rate $\lambda_i(t)$ through $\lambda_i(t) = F_i^{-1}(p_{c,i}(t))$ for all $i \in \mathcal{I}$ and the price $p_{s,j}(t)$ determines the arrival rate $\mu_j(t)$ through $\mu_j(t) = G_j^{-1}(p_{s,j}(t))$ for all $j \in \mathcal{J}$. After arrivals in the time slot $t$, the platform matches customers and servers in the queues via compatible links. Once a customer is matched with a server, both exit the system immediately.

**Objectives:** Consider a finite time horizon $T$. As in [1], we consider three performance metrics, profit, average queue length, and maximum queue length, as shown as follows:

$$\text{Profit}(T) = \sum_{t=1}^{T} \mathbb{E}\left[\sum_i \lambda_i(t) F_i(\lambda_i(t)) - \sum_j \mu_j(t) G_j(\mu_j(t))\right]$$

$$\text{AvgQLen}(T) = \frac{1}{T} \sum_{t=1}^{T} \mathbb{E}\left[\sum_i Q_{c,i}(t) + \sum_j Q_{s,j}(t)\right]$$

$$\text{MaxQLen}(T) = \max_{t=1,\dots,T} \max\left\{\max_i Q_{c,i}(t), \max_j Q_{s,j}(t)\right\}.$$

Our goal is to design an online pricing and matching algorithms to maximize the profit of the platform *without knowing the demand and supply functions*, while maintaining reasonable maximum and average queue lengths.

**Baseline:** We compare the profit of an online algorithm with the optimal value of a fluid-based optimization problem [2] as follows:

$$\max_{\boldsymbol{\lambda},\boldsymbol{\mu},\boldsymbol{x}} \sum_i \lambda_i F_i(\lambda_i) - \sum_j \mu_j G_j(\mu_j) \tag{1}$$

$$\text{s.t.} \quad \lambda_i = \sum_{j:(i,j)\in\mathcal{E}} x_{i,j} \text{ for all } i \in \mathcal{I}, \quad \mu_j = \sum_{i:(i,j)\in\mathcal{E}} x_{i,j} \text{ for all } j \in \mathcal{J}, \tag{2}$$

$$x_{i,j} \geqslant 0 \text{ for all } (i,j) \in \mathcal{E}, \quad \lambda_i, \mu_j \in [0, 1] \text{ for all } i \in \mathcal{I}, j \in \mathcal{J}, \tag{3}$$

where $\boldsymbol{\lambda} := (\lambda_i)_{i\in\mathcal{I}}$ and $\boldsymbol{\mu} := (\mu_j)_{j\in\mathcal{J}}$ can be viewed as steady-state arrival rates, and $\boldsymbol{x} := (x_{i,j})_{(i,j)\in\mathcal{E}}$ can be viewed as steady-state matching rates. The constraints (2) can be viewed as balance equations. We adopt the following assumptions as in [1, 2]:

**Assumption 2.** *The function $\lambda_i F_i(\lambda_i)$ is concave in $\lambda_i$ for all $i$ and the function $\mu_j G_j(\mu_j)$ is convex in $\mu_j$ for all $j$.*

**Assumption 3.** *There exists a known positive number $a_{\min} \in (0, 1)$ for which there exists an optimal solution $(\boldsymbol{\lambda}^*, \boldsymbol{\mu}^*, \boldsymbol{x}^*)$ to the optimization problem (1)-(3), satisfying $x_{i,j}^* > 0$, $a_{\min} \leqslant \lambda_i^* < 1$ and $a_{\min} \leqslant \mu_j^* < 1$ for all $i$ and $j$.*

Assumption 2 implies that the profit function $\lambda_i F_i(\lambda_i) - \mu_j G_j(\mu_j)$ is concave and the optimization problem defined by (1)-(3) is concave. This concavity assumption follows from the economic law of diminishing marginal return: as the arrival rate increases, the marginal return, that is, the derivative of the profit function, decreases, which implies that the profit function is concave. Assumption 3 means that there exists an optimal solution that resides in the interior of the feasible set. This means that no links or queues are redundant and no queues reach maximum arrival rates in this optimal solution. This is a mild assumption because:

- If any redundant links or queues exist, they can likely be easily identified in practice and removed from the problem.

- Suppose each time slot is set to be sufficiently small when modeling real-world arrival rates. Then, an arrival rate of 1 in this discrete-time model with Bernoulli arrivals corresponds to a sufficiently large actual arrival rate. In this case, no queue would reach the maximum arrival rate in the optimal solution due to the concavity of the profit function w.r.t. arrival rates.

We consider policies that make the queue mean rate stable [21], i.e., under the policy, for all $i, j$,

$$\lim_{T \to \infty} \frac{1}{T} \mathbb{E}[Q_{\mathrm{c},i}(T)] = 0, \qquad \lim_{T \to \infty} \frac{1}{T} \mathbb{E}[Q_{\mathrm{s},j}(T)] = 0. \tag{4}$$

As shown in [1], the optimal value of the fluid optimization problem (1)-(3) is an upper bound on the asymptotic time-averaged expected profit under any mean rate stable policy. Therefore, we use the optimal value of the fluid problem as baseline to define expected regret:

$$\mathbb{E}[R(T)] := T \left( \sum_i \lambda_i^* F_i(\lambda_i^*) - \sum_j \mu_j^* G_j(\mu_j^*) \right) - \sum_{t=1}^{T} \mathbb{E} \left[ \sum_i \lambda_i(t) F_i(\lambda_i(t)) - \sum_j \mu_j(t) G_j(\mu_j(t)) \right].$$
$$\tag{5}$$

**Challenges:** We face several challenges in addressing this problem:

- The first challenge is the *unknown* demand and supply functions $F_i$ and $G_j$. The objective and constraints of the optimization problem (1)-(3) are in terms of arrival rates but the platform can only control the prices directly. If we choose to use prices as the control variable of the optimization problem, then the problem becomes nonconcave due to nonlinear equations in the constraint set, and the constraint set becomes unknown. This is the key challenge compared to the case of the single-sided queue [19], where balance equations are not needed.

- We are considering an online learning setting where learning and decision making occur simultaneously. It is not acceptable to first estimate the demand and supply functions through running the system with difference prices (e.g., nonparametric regression) and then solve the optimization problem, because the queue lengths can grow linearly over time in the estimation process due to imbalanced arrival rates. An example can be found in Appendix A.

- It is challenging to characterize and achieve the optimal tradeoff between regret and queue length. The paper [1] obtain $\tilde{\Theta}(T^{1-\gamma})$ regret and $\tilde{\Theta}(T^{\gamma})$ average queue length bounds, but we know from [2] that this tradeoff is not optimal. It is worth studying whether there exists an algorithm that achieves a near-optimal tradeoff between regret and average queue length in this learning setting.

**Definition of Pareto-optimal tradeoff:** Let $\mathbb{E}_{\pi,d}[R(T)]$ denote the expected profit regret under policy $\pi$ and problem instance $d$. The problem instance $d$ specifies the bipartite graph $G(\mathcal{I} \cup \mathcal{J}, \mathcal{E})$ and the demand and supply functions. Let $\mathrm{AvgQLen}_{\pi,d}(T)$ denote the average queue length under policy $\pi$ and problem instance $d$. Let $(x(\pi), y(\pi))$ denote the exponents of the objective values under the worst case, i.e.,
$$x(\pi) := \limsup_{T \to \infty} \log_T (\sup_d \mathbb{E}_{\pi,d}[R(T)])$$
and
$$y(\pi) := \limsup_{T \to \infty} \log_T (\sup_d \mathrm{AvgQLen}_{\pi,d}(T)).$$
A point $(x(\pi_1), y(\pi_1))$ is said to be better than another point $(x(\pi_2), y(\pi_2))$ if $x(\pi_1) \leqslant x(\pi_2)$ and $y(\pi_1) \leqslant y(\pi_2)$ with at least one inequality strict. A point $(x(\pi), y(\pi))$ is Pareto-optimal if there is no other policy $\pi'$ that satisfies $x(\pi') \leqslant x(\pi)$ and $y(\pi') \leqslant y(\pi)$ with at least one inequality strict. The set of all Pareto-optimal points forms the Pareto frontier.

**Notation:** Bold symbols denote vectors in $\mathbb{R}^{|\mathcal{E}|}$, $\mathbb{R}^I$ or $\mathbb{R}^J$, such as $\boldsymbol{x}, \boldsymbol{\lambda}, \boldsymbol{\mu}$. Subscript $i$ or $j$ refer to individual elements of a vector; for example, $\lambda_i$ represents the $i^{\mathrm{th}}$ element of $\boldsymbol{\lambda}$. Note that $\boldsymbol{x} = (x_{i,j})_{(i,j) \in \mathcal{E}}$ is treated as a vector in $\mathbb{R}^{|\mathcal{E}|}$. The order of elements in $\boldsymbol{x}$ can be arbitrary, provided it is consistent. Additionally, we use the subscript c for the customer side and s for the server side.

## 3 Main Results

In this section, we provide an overview of the main results in this paper. We use a longest-queue-first matching algorithm and propose a novel learning-based pricing algorithm, which will be described in Section 4 in detail. The new algorithm achieves the following profit-regret bound and queue-length bounds:

$$\mathbb{E}[R(T)] = \tilde{O}\left(T^{1-\gamma}\right), \quad \mathrm{AvgQLen}(T) = \tilde{O}(T^{\frac{\gamma}{2}}), \quad \mathrm{MaxQLen}(T) = O\left(T^{\gamma}\right),$$

for any $\gamma \in [0, 1/6]$, where $\gamma$ is a hyperparameter in the proposed algorithm. The formal statements of the results can be found in Appendix B. As the allowable queue length increases, the achievable regret bound improves. However, the regret bound cannot be further reduced below $\tilde{O}(T^{5/6})$ in our result. Compared to [1], the average queue length in this tradeoff is significantly improved from $\tilde{O}(T^\gamma)$ to $\tilde{O}(T^{\gamma/2})$ while the regret and the maximum queue length remain the same.

**Near-optimal tradeoff:** This $\tilde{O}(T^{1-\gamma})$ versus $\tilde{O}(T^{\gamma/2})$ tradeoff between regret and average queue length matches the $\Theta(\eta^{1-\gamma})$ regret and $\Theta(\eta^{\gamma/2})$ queue length tradeoff in [2] up to logarithmic factors, where they show the tradeoff is optimal in their setting. The parameter $\eta$ in [2] is the scaling of arrival rates in their setting, which plays a similar role as the time horizon $T$ in our setting because both characterize the scaling of the number of arrivals to the system. We now establish that this trade-off between regret and average queue length is nearly optimal. We restrict ourselves to a single-link system, i.e., $I = J = 1$ and drop the subscripts $i$ and $j$. Consider the matching policy that matches all pairs whenever possible. Note that in a single-link system, there is no incentive to delay any possible match, as doing so only results in increased queue lengths. Similar to AvgQLen($T$), define AvgQ$^2$Len($T$) $:= \frac{1}{T} \sum_{t=1}^{T} \mathbb{E}[Q_c(t)^2 + Q_s(t)^2]$. Assuming an upper bound on AvgQ$^2$Len($T$), the lemma below obtains a lower bound on the regret.

**Lemma 1.** *Let Assumption 1 and Assumption 2 hold. Assume that the functions $-xF(x)$ (negative revenue) and $xG(x)$ (cost) are both strongly convex. Also assume the unique optimal solution to the fluid optimization problem $\lambda^* = \mu^* \notin \{0, 1\}$ to avoid the trivial case.*

*Fix a $\gamma \in [0, 1/2]$. Then, for sufficiently large $T$, for any pricing policy for which $\sqrt{AvgQ^2Len(T)} \leqslant T^{\gamma/2}$ and $\mathbb{E}[Q_c(T+1)^2 + Q_s(T+1)^2] = o(T)$, we must have $\mathbb{E}[R(T)] = \Omega(T^{1-\gamma})$.*

Proof of Lemma 1 can be found in Appendix J in [22]. Note that the above lemma assumes $\sqrt{\text{AvgQ}^2\text{Len}(T)} \leqslant T^{\gamma/2}$ which is slightly stronger than assuming AvgQLen($T$) $\leqslant T^{\gamma/2}$ as $\sqrt{\text{AvgQ}^2\text{Len}(T)} \geqslant$ AvgQLen($T$) by Jensen's inequality and the fact that $Q_c(t)^2 + Q_s(t)^2 = (Q_c(t) + Q_s(t))^2$ (by Lemma 1 in [1]). In particular, our proof strategy requires a handle on a higher moment of the queue length to obtain the required bound on the regret. In Appendix K in [22], we show that $\sqrt{\text{AvgQ}^2\text{Len}(T)} = \Theta(\text{AvgQLen}(T))$ for a large class of pricing policies that are a small perturbation of a static pricing policy satisfying a natural negative drift condition. Such a class of policies is ubiquitous in the literature [20, 2, 23] on dynamic pricing in queues.

To understand the $1 - \gamma$ versus $\gamma/2$ trade-off, consider a general pricing policy given by $\lambda(Q_c, Q_s), \mu(Q_c, Q_s) \in [\lambda^\star - \alpha, \lambda^\star + \alpha]$ for some $\alpha > 0$. For any such policy, the average queue length is at least $\Theta(1/\alpha)$ as the difference in the arrival rates is $\Theta(\alpha)$ [23]. Also, denote the regret as a function of the arrival rate for each time slot by $r$ and note that $r(\lambda^\star) = 0$. By Taylor's expansion, the cumulative regret in time $T$ for arrival rate $\lambda^\star \pm \Theta(\alpha)$ is $T\Theta(\alpha r'(\lambda^\star) + \alpha^2 r''(\lambda^\star)/2)$, where the first order term vanishes as $r'(\lambda^\star) \approx 0$ by the first order optimality conditions. Thus, in summary, the queue length is of the order $1/\alpha$ and the regret is $T\alpha^2$. By substituting $\alpha = T^{-\gamma/2}$, we obtain that the $1 - \gamma$ versus $\gamma/2$ trade-off is fundamental.

**Comparison with [1]:** The paper [1] employs a pricing policy with an admission control threshold $q^{\text{th}}$. In such a setting, the queue length is of the order $\Theta(q^{\text{th}})$ as the underlying system dynamics is a symmetric random walk between $[0, q^{\text{th}}]$. In addition, the regret is equal to $\sum_{t=1}^{T} \Pr[Q(t) = q^{\text{th}}]$, i.e., the proportion of the time we hit the boundary, which is approximately equal to $T/q^{\text{th}}$. Thus, by picking $q^{\text{th}} = T^\gamma$, it will result in a $1 - \gamma$ versus $\gamma$ trade-off. *This intuition shows that the approach of [1] will not be able to result in the optimal $1 - \gamma$ versus $\gamma/2$ trade-off.*

**Scaling with the number of customer/server types:** The regret upper bound scales with the number of customer types $I$ and server types $J$ as $\tilde{O}(I^4 J^4 (I + J) T^{1-\gamma})$. The average queue length upper bound scales with $I$ and $J$ as $\tilde{O}(IJ(I + J) T^{\gamma/2})$. The maximum queue length does not scale with $I$ or $J$. We believe the dependencies on $I$ and $J$ can be improved, as our analysis mainly focuses on obtaining the correct dependence in terms of the time horizon $T$.

# 4 Algorithm

In this section, we will describe the matching algorithm we use and propose a novel pricing algorithm, under which we can achieve the results in the previous section.

**Matching algorithm:** We use the same matching algorithm proposed in [1], which is a discretized version of the MaxWeight algorithm in [2]. At each time slot, we iterate through all queues on both the customer and server sides. For each queue, we first check for new arrivals. If a new arrival is detected (either a customer or a server), the algorithm matches it with an entity from the longest compatible queue on the opposite side.

**Pricing algorithm:** We propose a novel pricing algorithm that aims to achieve the optimal tradeoff between regret and average queue length, while having a guarantee of anytime maximum queue length. Same as [1], we use the techniques of zero-order stochastic projected gradient ascent [9] and bisection search in the pricing algorithm. The key innovation lies in how we implement the $\alpha$-perturbed two-price policy mentioned in the intuition in Section 3 in the *learning* setting. We propose a novel *probabilistic two-price policy*, where we reduce the arrival rate by perturbing the price with a constant probability for each queue when the queue is nonempty and the length is less than a predefined threshold. This *probabilistic two-price policy* achieves a near-optimal tradeoff between regret and average queue length, while having a guarantee of anytime maximum queue length. The rest of this section provides a detailed explanation of our pricing algorithm.

## 4.1 Two-Point Zero-Order Method

Note that under Assumption 3, the fluid problem (1)-(3) can be equivalently rewritten as

$$\max_{\boldsymbol{x}} f(\boldsymbol{x}) := \sum_i \left( \sum_{j \in \mathcal{E}_{c,i}} x_{i,j} \right) F_i \left( \sum_{j \in \mathcal{E}_{c,i}} x_{i,j} \right) - \sum_j \left( \sum_{i \in \mathcal{E}_{s,j}} x_{i,j} \right) G_j \left( \sum_{i \in \mathcal{E}_{s,j}} x_{i,j} \right) \tag{6}$$

$$\text{s.t.} \sum_{j \in \mathcal{E}_{c,i}} x_{i,j} \in [a_{\min}, 1] \text{ for all } i \in \mathcal{I}, \quad \sum_{i \in \mathcal{E}_{s,j}} x_{i,j} \in [a_{\min}, 1] \text{ for all } j \in \mathcal{J}, \tag{7}$$

$$x_{i,j} \geqslant 0 \text{ for all } (i, j) \in \mathcal{E}, \tag{8}$$

where $\mathcal{E}_{c,i} := \{j | (i, j) \in \mathcal{E}\}$ and $\mathcal{E}_{s,j} := \{i | (i, j) \in \mathcal{E}\}$. Given that the demand and supply functions $F_i$ and $G_j$ are unknown, we cannot solve the problem (6)-(8) directly. Suppose we have zero-order access to the objective function (6), i.e., the value of the objective function can be evaluated given any input variable $\boldsymbol{x}$. Then, we can use the two-point zero-order method [9]. We will present the main idea here. In the method, we begin with an initial feasible solution and perform projected gradient ascent, estimating the gradient at each iteration using two points that are perturbed from the current solution. Specifically, it can be described in the following steps:

**(1) Initial feasible point:** Let $\mathcal{D}$ denote the feasible set of the problem (6)-(8). Let $\mathcal{D}'$ denote a shrunk set of $\mathcal{D}$ with a parameter $\delta > 0$. The word "shrunk" means that $\boldsymbol{x} + \delta \boldsymbol{u} \in \mathcal{D}$ for any $\boldsymbol{x} \in \mathcal{D}'$ and any vector $\boldsymbol{u}$ in the unit ball, which was proved in [1] with the definition of $\mathcal{D}'$ under some mild assumptions. The details about $\mathcal{D}'$ are presented in Appendix C for completeness. The algorithm begins with an initial feasible solution $x(1) \in \mathcal{D}'$.

**(2) Generate a random direction and two points:** In each iteration $k$, we begin by sampling a unit vector $\boldsymbol{u}(k)$ in a uniformly random direction. Next, we perturb the current solution $\boldsymbol{x}(k)$ in the direction of $\boldsymbol{u}(k)$ and the opposite direction of $\boldsymbol{u}(k)$, generating two points, $\boldsymbol{x}(k) + \delta \boldsymbol{u}(k)$ and $\boldsymbol{x}(k) - \delta \boldsymbol{u}(k)$. We know that if the current solution $\boldsymbol{x}(k)$ is in the shrunk set $\mathcal{D}'$, then the two points will be in the feasible set $\mathcal{D}$ and hence are feasible.

**(3) Estimate the function values in the two points:** If we have zero-order access of the objective function $f$, we can obtain the function values $f(\boldsymbol{x}(k) + \delta \boldsymbol{u}(k))$ and $f(\boldsymbol{x}(k) - \delta \boldsymbol{u}(k))$. However, we do not have zero-order access to $f$ since we do not even have zero-order access to the demand and supply functions $F_i$ and $G_j$. In this case, we need to estimate the values of $F_i$ and $G_j$ given some known arrival rates as inputs. We will use a bisection search method similar to [1] to estimate the values, which will be described later in Section 4.2.

**(4) Gradient Calculation:** With the estimation of the two function values $\hat{f}(\boldsymbol{x}(k) + \delta \boldsymbol{u}(k))$ and $\hat{f}(\boldsymbol{x}(k) - \delta \boldsymbol{u}(k))$, the gradient $\hat{\boldsymbol{g}}(k) = \frac{|\mathcal{E}|}{2\delta} [\hat{f}(\boldsymbol{x}(k) + \delta \boldsymbol{u}(k)) - \hat{f}(\boldsymbol{x}(k) - \delta \boldsymbol{u}(k))] \boldsymbol{u}(k)$.

**(5) Projected gradient ascent:** With the estimated gradient $\hat{\boldsymbol{g}}(k)$, we perform a step of projected gradient ascent with a step size $\eta$: $\boldsymbol{x}(k + 1) = \Pi_{\mathcal{D}'}(\boldsymbol{x}(k) + \eta \hat{\boldsymbol{g}}(k))$.

Then the algorithm repeats Step (2)-(5) for the next iteration. This part of the pricing algorithm is the same as the balanced pricing algorithm in [1] and the details of the algorithm can be found in Appendix D for completeness.

## 4.2   Bisection Search Method

In this section, we will illustrate the bisection search method that we mentioned in the previous section. The bisection search takes the target arrival rates and the price searching intervals for all queues as inputs and outputs the estimated prices corresponding to the input arrival rates with accuracy $\epsilon$. The bisection search can be illustrated in the following steps:

**(1) Calculate the midpoints:** In each bisection iteration $m$ ($m = 1, 2, \ldots, M$, $M = \lceil \log_2(1/\epsilon) \rceil$), we first calculate the midpoint of the price searching interval for each queue. Note that the initial price searching intervals when $k = 1$ and $m = 1$ are predetermined at the beginning of the pricing algorithm, and the initial price searching intervals when $k > 1$ and $m = 1$ are calculated from the output of the bisection search in the previous outer iteration $k - 1$, which can be found in Line 11-17 in Algorithm 1. The underlying idea is that the price will not change too much since the corresponding target arrival rate will not change too much due to the small step size $\eta$.

**(2) Run the system to collect samples:** Next, we run the system with the midpoints for a certain number of time slots to obtain $N$ samples of each arrival rate corresponding to each midpoint, where $N = \lceil (\beta/\epsilon^2) \ln(1/\epsilon) \rceil$ and $\beta > 0$ is a constant. How we run the system to collect samples is the key difference between our proposed algorithm and that of [1]. We propose a *probabilistic two-price policy* and will present it later in Section 4.3.

**(3) Estimate the arrival rates and update the price searching intervals:** Then, we can obtain an estimate of the arrival rate for each queue by taking the average of $N$ samples. For each queue on the customer side, if the estimated arrival rate is greater than the input arrival rate, we should increase the price to reduce the arrival rate because the demand function is decreasing, so the price searching interval should be updated to the upper half of the previous price searching interval. Otherwise, the price searching interval should be updated to the lower half of the previous price searching interval. For each queue on the server side, we should do the opposite since the supply function is increasing.

Then the algorithm repeats Step (1)-(3) for the next bisection iteration. The details of the algorithm can be found in Appendix F.

## 4.3   Probabilistic Two-Price Policy

In this section, we propose a novel pricing policy called *probabilistic two-price policy* for Step (2) in Section 4.2. Recall that the goal of Step (2) is to collect samples to estimate the arrival rates corresponding to the prices of the midpoints. For an estimate of accuracy $\epsilon$, we need $N = \tilde{\Theta}(1/\epsilon^2)$ number of samples according to the central limit theorem.

**Challenges:** The key challenge in setting prices in this sample collection process is that we need to consider and balance three objectives. First, we need to set prices to be the midpoints to collect samples for learning. Second, we need to control the average queue length and maximum queue length during this sample collection process. Third, we need to minimize the regret. For fast learning, we want to set the prices to be the midpoints as frequently as possible. However, simply using the midpoints to run the system for $N$ time slots will not give us any way to control the queue lengths. To control the maximum queue length, we use a threshold $q^{\text{th}}$ as in [1] – when the queue length exceeds $q^{\text{th}}$, we reject arrivals by setting the highest price $p_{c,i,\max}$ for customer queue $i$ or the lowest price $p_{s,j,\min}$ for server queue $j$. However, this threshold policy cannot achieve an optimal tradeoff between *average queue length* and regret. A challenging question is whether we can achieve a near-optimal tradeoff.

We propose the *probabilistic two-price policy*, which resolves the tension between obtaining useful samples for fast learning and maintaining small queue lengths, providing a positive answer to the above question. The pseudo-code of the proposed policy can be found in Algorithm 3 in Appendix G. The proposed policy can be explained in the following steps:

(1) At the beginning of the algorithm, we initialize counters to track the number of useful arrival rate samples for each queue, as shown in Line 1 in Algorithm 3.

(2) Control the maximum queue length: As in [1], for each queue, we use a threshold $q^{\text{th}}$ to control the maximum queue length. If the queue length exceeds or equals the threshold, we will set the price to the maximum (for customer queues) or the minimum (for server queues) to reject new arrivals, as shown in Line 4 and Line 10 in Algorithm 3.

(3) **Control the average queue length with a probabilistic approach:** To control the average queue length, we propose a probabilistic approach of adjusting the price, to reduce the arrival rate for each queue when the queue is nonempty and the length is less than the threshold $q^{\text{th}}$. Specifically, for each customer queue, with a constant probability (set to $1/2$ for simplicity), we increase the price by a small amount $\alpha$ to reduce the arrival rate; for each server queue, with a constant probability, we decrease the price by $\alpha$ to reduce the arrival rate, as shown in Line 5 and Line 11 in Algorithm 3. When the queue is empty, we keep the original prices of the midpoints, as shown in Line 6 and Line 12.

(4) With the above prices, we run the system for one time slot. For each queue, we keep the arrival sample only when the original price is used, discarding the sample when the price is adjusted, as shown in Line 7 and Line 13 in Algorithm 3. We repeat Step (2) and Step (3) and run the system until the number of samples we keep is no less than $N$ for every queue. Finally, the algorithm returns the samples we keep.

**Difficulties introduced by learning and algorithmic innovations:** The paper [2] proposed a two-price policy (the $\alpha$-perturbed policy mentioned in Section 3) – when the queue length exceeds a predetermined threshold, they reduce the arrival rate by a small amount $\alpha$. The idea was to ensure that we always have an $\alpha$ negative drift that reduces the queue lengths in expectation. However, in the learning setting where the demand and supply functions are unknown, the objective of setting prices is not only to induce a negative drift to control queue lengths, but also to yield useful samples for learning. Note that every time we adjust the price by $\alpha$ to induce a negative drift, these arrival samples become biased. These biases influence the value estimates of the two points, which in turn affect the gradient estimates in the two-point gradient ascent method, ultimately impacting the regret. Using these biased samples will lead to a regret of at least order $\Theta(T\alpha)$. To achieve the optimal trade-off between average queue length and regret, the regret must be $\Theta(T\alpha^2)$ as mentioned in the intuition in Section 3. Therefore, in our policy, we discard these biased samples, making them unusable for learning. Hence, it is important to ensure that negative drift is enforced while not wasting too many samples. The novel probabilistic approach enables us to obtain useful samples with a constant fraction of time for fast learning, while also ensuring negative drift through reducing arrival rates by $\Theta(\alpha)$ with a constant probability when the queue is nonempty . By setting $\alpha$ we are able to achieve a near-optimal tradeoff between average queue length and regret.

**Technical innovations:** The analysis related to the proposed probabilistic two-price policy is novel. Specifically, the new technical challenges and the corresponding innovations include:

- One challenge in the analysis is to bound the number of useless samples that are collected at the time of adjusting prices to reduce the arrival rate for queue-length control. The proposed probabilistic approach controls the fraction of time spent adjusting prices, thereby bounding the number of useless samples. This can be formally established using Wald's lemma, as shown in Appendix L.4.1 in [22].

- Another challenge in the analysis is to bound the regret induced by adjusting prices by $\alpha$. The regret induced by adjusting prices can be bounded by the first-order error and the second-order error. The difficulty is to prove that the first-order error is no greater than the second-order error in the learning setting. To show this, we combine the KKT condition (Appendix L.3 in [22]) and the Lyapunov drift method (Lemma 5 in [22], proof in Appendix O in [22]).

### 4.4 Computational Complexity

The proposed pricing algorithm has computational complexity $\tilde{O}((I + J)T + IJT^{1-4\gamma})$, along with $T^{1-4\gamma}$ calls to a projection oracle. The projection can be implemented by solving a convex quadratic program using interior point methods, which have a computational complexity of $\tilde{O}((IJ)^{3.5})$ [24]. Hence, the overall computational complexity is $\tilde{O}((I + J)T + (IJ)^{3.5}T^{1-4\gamma})$.

# 5    Numerical Results

In this section, we present simulation results for the proposed algorithm and compare its performance with the *two-price policy* (with known demand and supply functions) from [2] and the *threshold policy* from [1].

To compare the performance of these algorithms, we consider the following objective function:

$$\sum_{\tau=1}^{t} \left( \sum_i \lambda_i(\tau) \left( p_{\text{c},i}(\tau) - w\mathbb{E}[W_{\text{c},i}(\tau)] \right) - \sum_j \mu_j(\tau) \left( p_{\text{s},j}(\tau) + w\mathbb{E}[W_{\text{s},j}(\tau)] \right) \right), \tag{9}$$

where $W_{\text{c},i}(\tau)$ denotes the waiting time of the customer arriving at queue $i$ at time slot $\tau$ and $W_{\text{s},j}(\tau)$ denotes the waiting time of the server arriving at queue $j$ at time slot $\tau$. The constant $w > 0$ can be viewed as the compensation the platform pays for per unit time the customer/server wait. This objective can be shown to be equivalent to maximizing the originally defined profit minus the sum of the queue lengths multiplied by $w$ (see Appendix H). Therefore, the regret of the objective (9) can be calculated by $\mathbb{E}[R(t)] + wt\text{AvgQLen}(t)$.

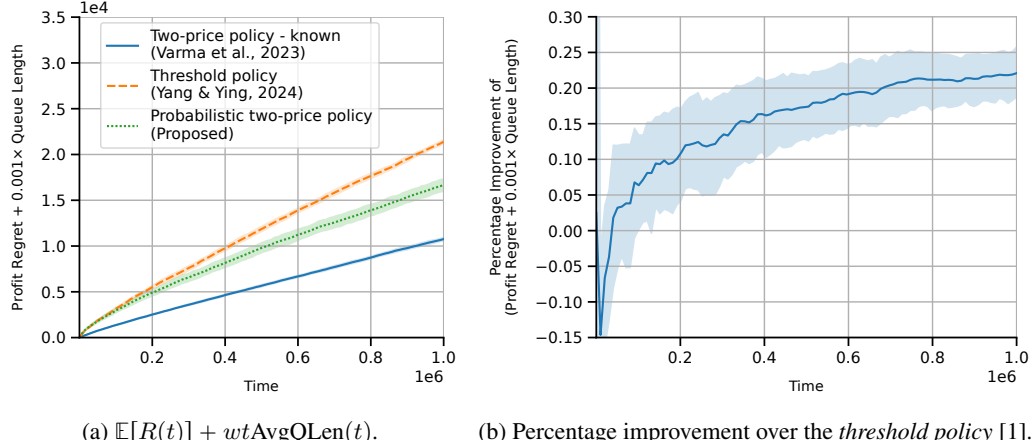

(a) $\mathbb{E}[R(t)] + wt\text{AvgQLen}(t)$.    (b) Percentage improvement over the *threshold policy* [1].

Figure 2: Comparison among the *two-price policy* (no learning, known demand and supply functions) [2], the *threshold policy* [1], and the proposed *probabilistic two-price policy* with $w = 0.001$. The shaded area is $95\%$ confidence interval with 10 independent runs.

Figure 2 presents the performance of the algorithms in a single-link system with $w = 0.001$ and $\gamma = 1/6$. We can see that the proposed algorithm significantly improves upon the algorithm in [1] by up to 22%. The *two-price policy* from [2] outperforms the other two algorithms because it is a genie-aided policy that assumes full knowledge of the true demand and supply functions, thereby bypassing the challenges associated with learning.

Details of the simulation are provided in Appendix H, where we report the regret, average queue length, and maximum queue length of these algorithms, and also compare their performance under a different weight value $w = 0.01$. We also test sensitivity of the performance of the proposed algorithm to parameter tuning. We conduct simulation for a multi-link system as well.

**Comparison with recent methods for queueing control:** Most of the work on reinforcement learning for queueing in the literature [25, 26, 27, 28, 16] does not utilize the special structure of our pricing and matching problem, and it is difficult to trade off between profit and queue length. When facing the curse of dimensionality of queueing systems, the works [25, 26, 28] use deep reinforcement learning, which does not have any theoretical guarantee on the performance. On the other hand, our approach circumvents the curse of dimensionality by restricting to the set of probabilistic two-price policies as opposed to a fully dynamic policy, as we know that a two-price policy is near-optimal. Other adaptive optimization approaches, such as those that discretize the price space and apply UCB-based methods, fail to balance matching rates and arrival rates on both sides, making it difficult to control queue lengths.

## Acknowledgments

The work of Zixian Yang and Lei Ying is supported in part by NSF under grants 2112471, 2207548, 2228974, and 2240981.

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

# A  An Estimate-Then-Optimize Algorithm

Consider the following estimate-then-optimize algorithm. First, we try prices uniformly in the price space. For each queue, to obtain an accuracy of $\zeta$ for the estimation of the demand/supply function, we need to discretize the space into $\Theta(1/\zeta)$ points, and for each point, we need at least $\Theta(1/\zeta^2)$ samples according to the central limit theorem. In this process, the queue length will increase linearly over time, up to $\Theta(1/\zeta^3)$. Using the estimated functions, we can obtain a solution with accuracy $\zeta$, and we can use this solution until the time horizon $T$. Hence, the worst-case average queue length will be $\Theta(1/\zeta^3 + (T - 1/\zeta^3)\zeta)$. The lowest worst-case average queue length that this approach can achieve is $\Theta(T^{3/4})$ with $\zeta = T^{-1/4}$, while our approach can achieve a much smaller average queue length $\tilde{\Theta}(T^{1/12})$ with $\gamma = 1/6$ according to the results in Section 3.

# B  Formal Statements of Theoretical Results

In this section, we will present formal theoretical results for the proposed algorithm described in Section 4, and discuss the difficulties in the analysis.

## B.1  Assumptions

To derive theoretical results, we impose the following additional assumptions.

We make the following assumption on the knowledge of the initial feasible point and the initial price searching intervals as in [1].

**Assumption 4** (Initial Balanced Arrival Rates). *Assume that the initial point $\boldsymbol{x}(1) \in \mathcal{D}'$ and the initial pricing searching intervals $\{[\underline{p}_{\mathrm{c},i}, \bar{p}_{\mathrm{c},i}], [\underline{p}_{\mathrm{s},j}, \bar{p}_{\mathrm{s},j}]\}_{i \in \mathcal{I}, j \in \mathcal{J}}$ satisfy*

$$\sum_{j \in \mathcal{E}_{\mathrm{c},i}} x_{i,j}(1) \in \left[ F_i^{-1}(\bar{p}_{\mathrm{c},i}) - \epsilon + \sqrt{|\mathcal{E}_{\mathrm{c},i}|}\delta, F_i^{-1}(\underline{p}_{\mathrm{c},i}) + \epsilon - \sqrt{|\mathcal{E}_{\mathrm{c},i}|}\delta \right], \textit{for all } i$$

$$\sum_{i \in \mathcal{E}_{\mathrm{s},j}} x_{i,j}(1) \in \left[ G_j^{-1}(\underline{p}_{\mathrm{s},j}) - \epsilon + \sqrt{|\mathcal{E}_{\mathrm{s},j}|}\delta, G_j^{-1}(\bar{p}_{\mathrm{s},j}) + \epsilon - \sqrt{|\mathcal{E}_{\mathrm{s},j}|}\delta \right], \textit{for all } j,$$

$$\bar{p}_{\mathrm{c},i} - \underline{p}_{\mathrm{c},i} \leqslant 2e_{\mathrm{c},i}, \quad \bar{p}_{\mathrm{s},j} - \underline{p}_{\mathrm{s},j} \leqslant 2e_{\mathrm{s},j}, \quad \textit{for all } i, j.$$

The small numbers $e_{\mathrm{c},i}$ and $e_{\mathrm{s},j}$ can be found in Appendix E.

**Discussion on Assumption 4:** As in [1], we assume that a feasible point $\boldsymbol{x}(1) \in \mathcal{D}'$ and the corresponding price bounds are known. We set the initial feasible point to be this point and use these bounds as the initial price searching intervals. Note that this point $\boldsymbol{x}(1)$ and the prices are usually not optimal. This assumption is mild, because in real application, we usually have some existing pricing strategy that may not be optimal but usually has approximately balanced arrival rates.

We make the following assumption about the objective function $f$ and an optimal solution.

**Assumption 5** (Weaker Strong-Convexity). *There exists a positive constant $\nu$ such that there is an optimal solution $(\boldsymbol{\lambda}^*, \boldsymbol{\mu}^*, \boldsymbol{x}^*)$ satisfying Assumption 3 and the following:*

$$f(\boldsymbol{x}^*) - f(\boldsymbol{x}) \geqslant \frac{\nu}{2} \|\boldsymbol{x} - \boldsymbol{x}^*\|_2^2, \tag{10}$$

*for all $\boldsymbol{x} \in \mathcal{D}'$.*

**Discussion on Assumption 5:** The equation (10) in Assumption 5 means that the optimal solution satisfies a property that is weaker than strong convexity. In fact, it is a necessary condition for strong convexity.

Under Assumption 3 and Assumption 5, without loss of generality, we can assume $a_{\min}$ satisfies

$$\sum_{j \in \mathcal{E}_{\mathrm{c},i}} \frac{a_{\min} + 1}{2N_{i,j}} - a_{\min} > 0 \quad \text{for all } i \tag{11}$$

$$\sum_{i \in \mathcal{E}_{\mathrm{s},j}} \frac{a_{\min} + 1}{2N_{i,j}} - a_{\min} > 0 \quad \text{for all } j \tag{12}$$

To see this, notice that (11) and (12) hold as long as the constant $a_{\min}$ is sufficiently small. If (11) or (12) does not hold, we can always find another constant $a'_{\min}$ such that $0 < a'_{\min} \leqslant a_{\min}$ and $a'_{\min}$ satisfies (11) and (12). Then, we can use this $a'_{\min}$ and all the results in this paper still hold.

We make the following assumption about the functions $F_i^{-1}$, $G_j^{-1}$, $F_i$, and $G_j$.

**Assumption 6** (Smoothness & Nonzero Derivatives). *We make the following assumptions on the demand and supply functions.*

*(1) Assume for all customer type $i$ and all server type $j$, the function $F_i^{-1}$ is $\beta_{F_i^{-1}}$-smooth, the function $G_j^{-1}$ is $\beta_{G_j^{-1}}$-smooth, the function $F_i$ is $\beta_{F_i}$-smooth, and the function $G_j$ is $\beta_{G_j}$-smooth, i.e.,*

$$\left| \left( \frac{\mathrm{d} F_i^{-1}(p)}{\mathrm{d}p} \right)_{p=p_1} - \left( \frac{\mathrm{d} F_i^{-1}(p)}{\mathrm{d}p} \right)_{p=p_2} \right| \leqslant \beta_{F_i^{-1}} |p_1 - p_2|,$$

$$\left| \left( \frac{\mathrm{d} G_j^{-1}(p)}{\mathrm{d}p} \right)_{p=p_1} - \left( \frac{\mathrm{d} G_j^{-1}(p)}{\mathrm{d}p} \right)_{p=p_2} \right| \leqslant \beta_{G_j^{-1}} |p_1 - p_2|,$$

$$\left| \left( \frac{\mathrm{d} F_i(\lambda)}{\mathrm{d}\lambda} \right)_{\lambda=\lambda_1} - \left( \frac{\mathrm{d} F_i(\lambda)}{\mathrm{d}\lambda} \right)_{\lambda=\lambda_2} \right| \leqslant \beta_{F_i} |\lambda_1 - \lambda_2|,$$

$$\left| \left( \frac{\mathrm{d} G_j(\mu)}{\mathrm{d}\mu} \right)_{\mu=\mu_1} - \left( \frac{\mathrm{d} G_j(\mu)}{\mathrm{d}\mu} \right)_{\mu=\mu_2} \right| \leqslant \beta_{G_j} |\mu_1 - \mu_2|$$

*for all prices $p_1, p_2$ and all arrival rates $\lambda_1, \lambda_2, \mu_1, \mu_2$.*

*(2) Assume for all customer type $i$ and all server type $j$, the derivatives of $F_i^{-1}$ and $G_j^{-1}$ are lower bounded by a constant, i.e.,*

$$\left| \frac{\mathrm{d} F_i^{-1}(p)}{\mathrm{d}p} \right| \geqslant C_{\mathrm{L}} \text{ for all } p \in [p_{\mathrm{c},i,\min}, p_{\mathrm{c},i,\max}],$$

$$\left| \frac{\mathrm{d} G_j^{-1}(p)}{\mathrm{d}p} \right| \geqslant C_{\mathrm{L}} \text{ for all } p \in [p_{\mathrm{s},j,\min}, p_{\mathrm{s},j,\max}],$$

*where $C_{\mathrm{L}}$ is a constant independent of $T$.*

**Discussion on Assumption 6:** Assumption 6(1) means that the demand and supply functions $F_i^{-1}$, $G_j^{-1}$, $F_i$, and $G_j$ are smooth. The smoothness assumption is common in the optimization literature. Assumption 6(2) means that the derivatives of these functions are bounded away from zero, which is necessary to ensure that the arrival rate can be reduced through adjusting the price. Nonzero derivatives is usually true in practice.

## B.2 Theoretical Bounds

Suppose we use the proposed algorithm described in Section 4. Let the parameters $\eta$, $\epsilon$, $\delta$, $\alpha$, $q^{\mathrm{th}}$ be functions of $T$. Let $\eta$, $\epsilon$, $\delta$, $\alpha$ be nonincreasing in $T$ and $q^{\mathrm{th}}$ be nondecreasing in $T$. We have the following theoretical bounds.

**Theorem 1.** *Let Assumption 1, 2, 3, 4, 5, 6 hold. Suppose $\epsilon < \delta$, $T \geqslant 2MN$, $\alpha$ is orderwise greater than $\left( \frac{\eta\epsilon}{\delta} + \eta + \delta + \epsilon \right)$, i.e., $\lim_{T\to\infty} \left( \frac{\eta\epsilon}{\delta} + \eta + \delta + \epsilon \right)/\alpha = 0$, then for sufficiently large $T$, we have*

$$\mathbb{E}[R(T)] = O\left( \frac{T}{q^{\mathrm{th}}} + \frac{\log^2(1/\epsilon)}{\eta\epsilon^2} + T\left( \frac{\epsilon}{\delta} + \eta + \delta \right) + \frac{\alpha\sqrt{T}\log(1/\epsilon)}{\epsilon\sqrt{\eta}} + T\alpha\left( \sqrt{\eta} + \sqrt{\frac{\epsilon}{\delta}} + \sqrt{\delta} \right) \right.$$

$$\left. + T\alpha^2 + q^{\mathrm{th}} + T^2\epsilon^{\frac{\beta}{2}+1} + T\epsilon^{\frac{\beta}{2}-1}\log(1/\epsilon) \right), \tag{13}$$

*and the queue lengths*

$$AvgQLen(T) \leqslant \Theta\left( \frac{1}{\alpha} + \frac{q^{\mathrm{th}}}{\alpha}\left( T\epsilon^{\frac{\beta}{2}+1} + \epsilon^{\frac{\beta}{2}-1}\log(1/\epsilon) \right) \right), \quad MaxQLen(T) \leqslant q^{\mathrm{th}}. \tag{14}$$

Proof of Theorem 1 can be found in Appendix L in [22]. In the regret bound (13), the term $T/q^{\text{th}}$ is caused by rejecting arrivals when queue lengths exceed $q^{\text{th}}$. The term $\log^2(1/\epsilon)/(\eta\epsilon^2)+T(\epsilon/\delta+\eta+\delta)$ is caused by the zero-order projected stochastic gradient ascent with biased gradient estimation. The term $\alpha\sqrt{T}\log(1/\epsilon)/(\epsilon\sqrt{\eta}) + T\alpha(\sqrt{\eta} + \sqrt{\frac{\epsilon}{\delta}} + \sqrt{\delta}) + T\alpha^2 + q^{\text{th}}$ is caused by reducing arrival rates through adjusting the prices by $\alpha$. The term $T^2\epsilon^{\frac{\beta}{2}+1} + T\epsilon^{\frac{\beta}{2}-1}\log(1/\epsilon)$ is caused by the estimation error of the arrival rates.

In order to obtain the tradeoff among regret, average queue length, and maximum queue length, we first set $q^{\text{th}} = T^\gamma$ for a fixed $\gamma$. By first optimizing the order of the regret bound (13) with respect to the parameters $\epsilon, \eta, \delta, \alpha$, and then optimizing the order of the average queue length bound (14) over $\alpha$ without compromising the order of the regret bound, we can obtain the following corollary.

**Corollary 1.** *Let all the assumptions in Theorem 1 hold. For any $\gamma \in (0, \frac{1}{6}]$, setting parameters $q^{\text{th}} = T^\gamma$, $\epsilon = T^{-2\gamma}$, $\eta = \delta = T^{-\gamma}$, $\alpha = T^{-\gamma/2}$, $\beta = 1/\gamma - 1$, for sufficiently large T, we can achieve a sublinear regret as*

$$\mathbb{E}[R(T)] = \tilde{O}(T^{1-\gamma}),$$

*and the queue lengths*

$$AvgQLen(T) = \tilde{O}(T^{\frac{\gamma}{2}}), \quad MaxQLen(T) \leqslant T^\gamma.$$

*For any $\gamma \in (\frac{1}{6}, 1]$, setting parameters $q^{\text{th}} = T^\gamma$, $\epsilon = T^{-1/3}$, $\eta = \delta = T^{-1/6}$, $\alpha = T^{-1/12}$, $\beta = 5$, for sufficiently large T, we can achieve a sublinear regret as*

$$\mathbb{E}[R(T)] = \tilde{O}(T^{5/6}),$$

*and the queue lengths*

$$AvgQLen(T) = \tilde{O}(T^{1/12}), \quad MaxQLen(T) \leqslant T^\gamma.$$

We can see from Corollary 1 that compared to [1], the average queue length bound is significantly improved from $\tilde{O}(T^\gamma)$ to $\tilde{O}(T^{\frac{\gamma}{2}})$ with the use of the *probabilistic two-price policy* while the regret bound remains the same order, . We remark that this tradeoff between regret and average queue length, $\tilde{O}(T^{1-\gamma})$ versus $\tilde{O}(T^{\frac{\gamma}{2}})$, matches the optimal tradeoff in [2] up to logarithmic order, although they do not consider learning in their setting. As the allowable maximum queue length increases (up to $T^{1/6}$), the achievable regret bound improves. However, increasing the maximum queue length beyond $T^{1/6}$ does not further enhance the regret bound, which remains at $\tilde{O}(T^{5/6})$ (after parameter optimization). Hence, we will not let $\gamma$ increase over $1/6$.

### B.3 Discussion

**Reason for choosing a probabilistic approach:** The probabilistic pricing scheme adjusts the price to reduce the arrival rate with a certain probability for each nonempty queue. Therefore, for each nonempty queue, with a certain probability, the arrival rate will be strictly smaller than the sum of the average matching rates over all the connected links. So the lengths of nonempty queues will decrease, making the average queue length bounded. Other approaches, such as threshold policies in [2], can also control the average queue length, but they will induce a larger regret or do not have any theoretical guarantee for the induced regret. The reason is that these threshold approaches introduce strong correlation among price adjustment decisions across different time slots, making it difficult to bound the number of useless samples. The proposed probabilistic pricing decouples this correlation, providing theoretically guaranteed regret and average queue length, as well as a near-optimal tradeoff between them.

**Range of $\gamma$:** One (possible) limitation of our result is the restricted range of $\gamma \in (0, 1/6]$. First, note that, in the learning setting, we cannot expect to have a better regret than $\sqrt{T}$ as shown in [29]. Thus, we should expect to have at best $\gamma \in (0, 1/2]$. In our setting, we incur an even larger regret of $T^{5/6}$ as the function we are maximizing is a concave function of the arrival rates but we cannot directly set the arrival rates; we are only allowed to set the prices which in turn determines the arrival rates. This extra dependency forces us to implement a line search to first try several different prices to set the correct arrival rate, which in turn allows us to implement one iteration of the gradient algorithm. This

additional step seems to be necessary due to the constraints imposed by the system and results in an additional regret. Having said that, we are not sure what the best possible regret is (somewhere between $\sqrt{T}$ and $T^{5/6}$), and it is an interesting future direction. We believe such a lower bound is non-trivial due to the dual learning setup (using prices to learn the arrival rate and then using the arrival rate to learn the optimal profit).

**Possible Extension of the Probabilistic Approach:** In problems of joint queueing, learning, and optimization, the correlation among them is usually complicated. One of the fundamental trade-offs in such problems is to optimize an objective while ensuring sustained low queue lengths. Thus, making mistakes during online learning in queues has a sustained effect as queues build up, and so, one needs to carefully select the actions (prices in our case) to determine the optimal action while keeping low queue lengths. The idea of a probabilistic algorithm partially decouples these two phenomena: it allows to exploration of the prices to optimize the profit while ensuring low queue lengths. This decoupling allows for a tight analysis, resulting in characterizing the optimal trade-off between profit and queue length. Thus, we believe that the idea of using such a probabilistic algorithm could be widely applicable in the context of online learning in queues.

## C  Definitions of the Shrunk Feasible Set $\mathcal{D}'$

The definition of the shrunk set $\mathcal{D}'$ is as follows.

$$\mathcal{D}' := \left\{ \boldsymbol{x} \,\middle|\, \text{for all } i, j, \; x_{i,j} - \frac{a_{\min}+1}{2N_{i,j}} \geqslant -\left(1 - \frac{\delta}{r}\right)\frac{a_{\min}+1}{2N_{i,j}}, \right.$$

$$\sum_{j'\in\mathcal{E}_{\mathrm{c},i}} \left( x_{i,j'} - \frac{a_{\min}+1}{2N_{i,j'}} \right) \in \left[ -\left(1 - \frac{\delta}{r}\right)\left( \sum_{j'\in\mathcal{E}_{\mathrm{c},i}} \frac{a_{\min}+1}{2N_{i,j'}} - a_{\min} \right), \right.$$

$$\left(1 - \frac{\delta}{r}\right)\left(1 - \sum_{j'\in\mathcal{E}_{\mathrm{c},i}} \frac{a_{\min}+1}{2N_{i,j'}}\right)\right],$$

$$\sum_{i'\in\mathcal{E}_{\mathrm{s},j}} \left( x_{i',j} - \frac{a_{\min}+1}{2N_{i',j}} \right) \in \left[ -\left(1 - \frac{\delta}{r}\right)\left( \sum_{i'\in\mathcal{E}_{\mathrm{s},j}} \frac{a_{\min}+1}{2N_{i',j}} - a_{\min} \right), \right.$$

$$\left. \left.\left(1 - \frac{\delta}{r}\right)\left(1 - \sum_{i'\in\mathcal{E}_{\mathrm{s},j}} \frac{a_{\min}+1}{2N_{i',j}}\right)\right]\right\}, \tag{15}$$

where $N_{i,j} := \max\{|\mathcal{E}_{\mathrm{c},i}|, |\mathcal{E}_{\mathrm{s},j}|\}$ denotes the maximum cardinality of the sets $\mathcal{E}_{\mathrm{c},i}$ and $\mathcal{E}_{\mathrm{s},j}$,

$$r := \min_{i,j}\left\{ \frac{1+a_{\min}}{2N_{i,j}}, \; \frac{1}{|\mathcal{E}_{\mathrm{c},i}|}\left(1 - \sum_{j'\in\mathcal{E}_{\mathrm{c},i}} \frac{a_{\min}+1}{2N_{i,j'}}\right), \; \frac{1}{|\mathcal{E}_{\mathrm{s},j}|}\left(1 - \sum_{i'\in\mathcal{E}_{\mathrm{s},j}} \frac{a_{\min}+1}{2N_{i',j}}\right), \right.$$

$$\left. \frac{1}{|\mathcal{E}_{\mathrm{c},i}|}\left( \sum_{j'\in\mathcal{E}_{\mathrm{c},i}} \frac{a_{\min}+1}{2N_{i,j'}} - a_{\min} \right), \; \frac{1}{|\mathcal{E}_{\mathrm{s},j}|}\left( \sum_{i'\in\mathcal{E}_{\mathrm{s},j}} \frac{a_{\min}+1}{2N_{i',j}} - a_{\min} \right) \right\}, \tag{16}$$

and $\delta \in (0, r)$.

Under the conditions (11) and (12), it can be shown that $\boldsymbol{x} + \delta\boldsymbol{u} \in \mathcal{D}$ for any $\boldsymbol{x} \in \mathcal{D}'$ and any vector $\boldsymbol{u}$ in the unit ball [1].

# D   Pricing Algorithm – Two-Point Zero-Order Projected Gradient Ascent

We present the details of the pricing algorithm – two-point zero-order projected gradient ascent in Algorithm 1, which is the same as the balanced pricing algorithm in [1]. The index $(k)$ denotes the $k^{\text{th}}$ outer iteration (iteration of the gradient ascent) and $(k, m)$ denotes the $m^{\text{th}}$ bisection iteration in the $k^{\text{th}}$ outer iteration.

---

**Algorithm 1** Pricing algorithm

---

1: **Initialize:** Choose an exploration parameter $\delta \in (0, r)$. Choose a step size $\eta \in (0, 1)$. Choose an accuracy parameter $\epsilon \in (0, 1/e)$. Choose a queue length threshold $q^{\text{th}}$. Choose a two-price parameter $\alpha$. Choose $\boldsymbol{x}(1) \in \mathcal{D}'$ as the initial point. Choose some initial price searching intervals $[\underline{p}_{c,i}, \bar{p}_{c,i}]$ for customer type $i$, $[\underline{p}_{s,j}, \bar{p}_{s,j}]$ for server type $j$. Define $e_{c,i}$ and $e_{s,j}$ according to (17) and (18) in Appendix E. Define $N := \left\lceil \frac{\beta \ln(1/\epsilon)}{\epsilon^2} \right\rceil$ and $M := \left\lceil \log_2 \frac{1}{\epsilon} \right\rceil$, where $\beta > 0$ is a constant.

2: Time step counter $t \leftarrow 1$

3: Outer iteration counter $k \leftarrow 1$

4: **repeat**

5:     **// generate a random direction and two points**

6:     Choose a unit vector $\boldsymbol{u}(k) \in \mathbb{R}^{|\mathcal{E}|}$ uniformly at random, i.e., $\|\boldsymbol{u}(k)\|_2 = 1$

7:     Let $x_{i,j}^+(k) := (\boldsymbol{x}(k) + \delta \boldsymbol{u}(k))_{i,j}$ and $x_{i,j}^-(k) := (\boldsymbol{x}(k) - \delta \boldsymbol{u}(k))_{i,j}$

8:     Let $\lambda_i^+(k) := \sum_{j \in \mathcal{E}_{c,i}} x_{i,j}^+(k)$, $\lambda_i^-(k) := \sum_{j \in \mathcal{E}_{c,i}} x_{i,j}^-(k)$, $\mu_j^+(k) := \sum_{i \in \mathcal{E}_{s,j}} x_{i,j}^+(k)$, $\mu_j^-(k) := \sum_{i \in \mathcal{E}_{s,j}} x_{i,j}^-(k)$

9:     Let $\boldsymbol{\lambda}^+(k)$ be a vector of $\lambda_i^+(k), i = 1 \ldots, I$. Define similarly $\boldsymbol{\lambda}^-(k)$, $\boldsymbol{\mu}^+(k)$, and $\boldsymbol{\mu}^-(k)$

10:     **// bisection search to approximate $F_i(\lambda_i^+(k)), F_i(\lambda_i^-(k)), G_j(\mu_j^+(k)), G_j(\mu_j^-(k))$ to estimate the profits in the two points.**

11:     **if** $k = 1$ **then**

12:         For all $i$, let $\underline{p}_{c,i}^+(k, 1) = \underline{p}_{c,i}, \bar{p}_{c,i}^+(k, 1) = \bar{p}_{c,i}$

13:         For all $j$, let $\underline{p}_{s,j}^+(k, 1) = \underline{p}_{s,j}, \bar{p}_{s,j}^+(k, 1) = \bar{p}_{s,j}$

14:     **else**

15:         For all $i$, let $\underline{p}_{c,i}^+(k, 1) = p_{c,i}^+(k - 1, M) - e_{c,i}, \bar{p}_{c,i}^+(k, 1) = p_{c,i}^+(k - 1, M) + e_{c,i}$

16:         For all $j$, let $\underline{p}_{s,j}^+(k, 1) = p_{s,j}^+(k - 1, M) - e_{s,j}, \bar{p}_{s,j}^+(k, 1) = p_{s,j}^+(k - 1, M) + e_{s,j}$

17:     **end if**

18:     Let $\underline{\boldsymbol{p}}_c^+(k, 1)$ be a vector of $\underline{p}_{c,i}^+(k, 1), i = 1 \ldots, I$. Define similarly $\bar{\boldsymbol{p}}_c^+(k, 1)$, $\underline{\boldsymbol{p}}_s^+(k, 1)$, $\bar{\boldsymbol{p}}_s^+(k, 1)$.

19:     $t, \boldsymbol{p}_c^+(k, M), \boldsymbol{p}_s^+(k, M) =$

20:             $\text{Bisection}\Big(\boldsymbol{\lambda}^+(k), \boldsymbol{\mu}^+(k), \underline{\boldsymbol{p}}_c^+(k, 1), \bar{\boldsymbol{p}}_c^+(k, 1), \underline{\boldsymbol{p}}_s^+(k, 1), \bar{\boldsymbol{p}}_s^+(k, 1), \alpha, q^{\text{th}}, t, M, N, \epsilon\Big)$

21:     Do Line 11-20 for $\boldsymbol{\lambda}^-(k), \boldsymbol{\mu}^-(k)$. Denote the counterparts of the prices by $\underline{\boldsymbol{p}}_c^-(k, 1), \bar{\boldsymbol{p}}_c^-(k, 1)$, $\boldsymbol{p}_c^-(k, M), \underline{\boldsymbol{p}}_s^-(k, 1), \bar{\boldsymbol{p}}_s^-(k, 1), \boldsymbol{p}_s^-(k, M)$.

22:     **// gradient calculation**

23:     Let $\hat{\boldsymbol{g}}(k) = \frac{|\mathcal{E}|}{2\delta} \Bigg[ \Big(\sum_{i=1}^I \lambda_i^+(k) p_{c,i}^+(k, M) - \sum_{j=1}^J \mu_j^+(k) p_{s,j}^+(k, M)\Big)$

24:             $- \Big(\sum_{i=1}^I \lambda_i^-(k) p_{c,i}^-(k, M) - \sum_{j=1}^J \mu_j^-(k) p_{s,j}^-(k, M)\Big) \Bigg] \boldsymbol{u}(k)$

25:     **// gradient ascent update**

26:     Projected Gradient Ascent: $\boldsymbol{x}(k + 1) = \Pi_{\mathcal{D}'}(\boldsymbol{x}(k) + \eta \hat{\boldsymbol{g}}(k))$;

27:     $k \leftarrow k + 1$;

28: **until** $t > T$

---

# E  Definitions of $e_{\mathrm{c},i}$ and $e_{\mathrm{s},j}$

The definitions of $e_{\mathrm{c},i}$ and $e_{\mathrm{s},j}$ are shown as follows:

$$
\begin{aligned}
e_{\mathrm{c},i} =& \frac{2\eta\epsilon|\mathcal{E}|^{3/2}L_{F_i}}{\delta}\left[\sum_{i'=1}^{I} L_{F_{i'}}\left(1 + L_{F_{i'}^{-1}}\left(p_{\mathrm{c},i',\max} - p_{\mathrm{c},i',\min}\right)\right)\right. \\
& + \left.\sum_{j'=1}^{J} L_{G_{j'}}\left(1 + L_{G_{j'}^{-1}}\left(p_{\mathrm{s},j',\max} - p_{\mathrm{s},j',\min}\right)\right)\right] + 2\epsilon L_{F_i}\left(1 + L_{F_i^{-1}}\left(p_{\mathrm{c},i,\max} - p_{\mathrm{c},i,\min}\right)\right) \\
& + \eta|\mathcal{E}|^{3/2}L_{F_i}\left(\sum_{i'=1}^{I}|\mathcal{E}_{\mathrm{c},i'}|(L_{F_{i'}} + p_{\mathrm{c},i',\max}) + \sum_{j'=1}^{J}|\mathcal{E}_{\mathrm{s},j'}|(L_{G_{j'}} + p_{\mathrm{s},j',\max})\right) + 2\delta|\mathcal{E}|^{1/2}L_{F_i}
\end{aligned}
\tag{17}
$$

$$
=\Theta\left(\frac{\eta\epsilon}{\delta} + \eta + \delta + \epsilon\right),
$$

$$
\begin{aligned}
e_{\mathrm{s},j} =& \frac{2\eta\epsilon|\mathcal{E}|^{3/2}L_{G_j}}{\delta}\left[\sum_{i'=1}^{I} L_{F_{i'}}\left(1 + L_{F_{i'}^{-1}}\left(p_{\mathrm{c},i',\max} - p_{\mathrm{c},i',\min}\right)\right)\right. \\
& + \left.\sum_{j'=1}^{J} L_{G_{j'}}\left(1 + L_{G_{j'}^{-1}}\left(p_{\mathrm{s},j',\max} - p_{\mathrm{s},j',\min}\right)\right)\right] + 2\epsilon L_{G_j}\left(1 + L_{G_j^{-1}}\left(p_{\mathrm{s},j,\max} - p_{\mathrm{s},j,\min}\right)\right) \\
& + \eta|\mathcal{E}|^{3/2}L_{G_j}\left(\sum_{i'=1}^{I}|\mathcal{E}_{\mathrm{c},i'}|(L_{F_{i'}} + p_{\mathrm{c},i',\max}) + \sum_{j'=1}^{J}|\mathcal{E}_{\mathrm{s},j'}|(L_{G_{j'}} + p_{\mathrm{s},j',\max})\right) + 2\delta|\mathcal{E}|^{1/2}L_{G_j}
\end{aligned}
\tag{18}
$$

$$
=\Theta\left(\frac{\eta\epsilon}{\delta} + \eta + \delta + \epsilon\right).
$$

# F   Bisection Search

We present the details of the bisection search method in Algorithm 2, which is similar to that in [1]. The key difference over [1] is how we run the system to collect samples. The inputs to the bisection search include the arrival rates $\boldsymbol{\lambda}^{+/-}(k)$ and $\boldsymbol{\mu}^{+/-}(k)$, the price searching intervals $\underline{\boldsymbol{p}}_{\mathrm{c}}^{+/-}(k,1), \bar{\boldsymbol{p}}_{\mathrm{c}}^{+/-}(k,1), \underline{\boldsymbol{p}}_{\mathrm{s}}^{+/-}(k,1), \bar{\boldsymbol{p}}_{\mathrm{s}}^{+/-}(k,1)$, the two-price parameter $\alpha$, the threshold parameter $q^{\mathrm{th}}$, the number of bisection iterations $M$, and the number of samples for each queue in each bisection iteration $N$. The bisection search will output the prices $\boldsymbol{p}_{\mathrm{c}}^{+/-}(k,M)$ and $\boldsymbol{p}_{\mathrm{s}}^{+/-}(k,M)$ such that $\lambda_i^{+/-}(k) \approx F_i^{-1}(p_{\mathrm{c},i}^{+/-}(k,M))$ and $\mu_j^{+/-}(k) \approx G_j^{-1}(p_{\mathrm{s},j}^{+/-}(k,M))$ for all $i, j$.

---

**Algorithm 2** Bisection $\left(\boldsymbol{\lambda}^{+/-}(k), \boldsymbol{\mu}^{+/-}(k), \underline{\boldsymbol{p}}_{\mathrm{c}}^{+/-}(k,1), \bar{\boldsymbol{p}}_{\mathrm{c}}^{+/-}(k,1), \underline{\boldsymbol{p}}_{\mathrm{s}}^{+/-}(k,1), \bar{\boldsymbol{p}}_{\mathrm{s}}^{+/-}(k,1), \alpha, q^{\mathrm{th}}, t, M, N, \epsilon\right)$

---

1: **for** $m = 1$ to $M$ **do**
2:   **// Calculate the midpoints**
3:   Let $p_{\mathrm{c},i}^{+/-}(k,m) = \frac{1}{2}\left(\underline{p}_{\mathrm{c},i}^{+/-}(k,m) + \bar{p}_{\mathrm{c},i}^{+/-}(k,m)\right)$ for all $i$
4:   Let $p_{\mathrm{s},j}^{+/-}(k,m) = \frac{1}{2}\left(\underline{p}_{\mathrm{s},j}^{+/-}(k,m) + \bar{p}_{\mathrm{s},j}^{+/-}(k,m)\right)$ for all $j$
5:   **// Run the system to collect $N$ samples with the midpoints**
6:   Let $t_{\mathrm{c},i}^{+/-}(k,m,n)$ denote the time slot when the price $p_{\mathrm{c},i}^{+/-}(k,m)$ is run for the $n^{\mathrm{th}}$ time for the customer queue $i$; Let $t_{\mathrm{s},j}^{+/-}(k,m,n)$ denote the time slot when the price $p_{\mathrm{s},j}^{+/-}(k,m)$ is run for the $n^{\mathrm{th}}$ time for the server queue $j$
7:   $t, (A_{\mathrm{c},i}(t_{\mathrm{c},i}^{+/-}(k,m,n)))_{n\in[N],i\in\mathcal{I}}, (A_{\mathrm{s},j}(t_{\mathrm{s},j}^{+/-}(k,m,n)))_{n\in[N],j\in\mathcal{J}}$
   $= \mathrm{ProbTwoPrice}\left(\boldsymbol{p}_{\mathrm{c}}^{+/-}(k,m), \boldsymbol{p}_{\mathrm{s}}^{+/-}(k,m), \alpha, q^{\mathrm{th}}, N, t\right),$ **// Key innovation over [1]**
   where $\boldsymbol{p}_{\mathrm{c}}^{+/-}(k,m)$ denotes a vector of $p_{\mathrm{c},i}^{+/-}(k,M), i \in \mathcal{I}$ and similarly for $\boldsymbol{p}_{\mathrm{s}}^{+/-}(k,m)$.
8:   **for** $i = 1$ to $I$ **do**
9:     **// Estimate the arrival rates**
10:     Let $\hat{\lambda}_i^{+/-}(k,m) = \frac{1}{N}\sum_{n=1}^{N} A_{\mathrm{c},i}(t_{\mathrm{c},i}^{+/-}(k,m,n))$ **// sample average**
11:     **// Update the price searching intervals**
12:     **if** $\hat{\lambda}_i^{+/-}(k,m) > \lambda_i^{+/-}(k)$ **then**
13:       $\underline{p}_{\mathrm{c},i}^{+/-}(k,m+1) = p_{\mathrm{c},i}^{+/-}(k,m), \bar{p}_{\mathrm{c},i}^{+/-}(k,m+1) = \bar{p}_{\mathrm{c},i}^{+/-}(k,m)$
14:     **else**
15:       $\underline{p}_{\mathrm{c},i}^{+/-}(k,m+1) = \underline{p}_{\mathrm{c},i}^{+/-}(k,m), \bar{p}_{\mathrm{c},i}^{+/-}(k,m+1) = p_{\mathrm{c},i}^{+/-}(k,m)$
16:     **end if**
17:   **end for**
18:   **for** $j = 1$ to $J$ **do**
19:     **// Estimate the arrival rates**
20:     Let $\hat{\mu}_j^{+/-}(k,m) = \frac{1}{N}\sum_{n=1}^{N} A_{\mathrm{s},j}(t_{\mathrm{s},j}^{+/-}(k,m,n))$ **// sample average**
21:     **// Update the price searching intervals**
22:     **if** $\hat{\mu}_j^{+/-}(k,m) > \mu_j^{+/-}(k)$ **then**
23:       $\underline{p}_{\mathrm{s},j}^{+/-}(k,m+1) = \underline{p}_{\mathrm{s},j}^{+/-}(k,m), \bar{p}_{\mathrm{s},j}^{+/-}(k,m+1) = p_{\mathrm{s},j}^{+/-}(k,m)$
24:     **else**
25:       $\underline{p}_{\mathrm{s},j}^{+/-}(k,m+1) = p_{\mathrm{s},j}^{+/-}(k,m), \bar{p}_{\mathrm{s},j}^{+/-}(k,m+1) = \bar{p}_{\mathrm{s},j}^{+/-}(k,m)$
26:     **end if**
27:   **end for**
28: **end for**
29: **Return** $t, \boldsymbol{p}_{\mathrm{c}}^{+/-}(k,M), \boldsymbol{p}_{\mathrm{s}}^{+/-}(k,M)$

---

# G    Probabilistic Two-Price Policy

The pseudo-code of the proposed probabilistic two-price policy is presented in Algorithm 3, where $t_{\mathrm{c},i}^{+/-}(k,m,n)$ denotes the time slot in which the price of the midpoint is run for the $n^{\mathrm{th}}$ time for the customer queue $i$ and $t_{\mathrm{s},j}^{+/-}(k,m,n)$ for server queue $j$.

---

**Algorithm 3** ProbTwoPrice$\big(\boldsymbol{p}_{\mathrm{c}}^{+/-}(k,m),\boldsymbol{p}_{\mathrm{s}}^{+/-}(k,m),\alpha,q^{\mathrm{th}},N,t\big)$

---

1: Let $n_{\mathrm{c},i}(k,m)=0$ and $n_{\mathrm{s},j}(k,m)=0$ for all $i,j$.
2: **repeat**
3:     **for** $i=1$ to $I$ **do**
4:         **if** $Q_{\mathrm{c},i}(t)\geqslant q^{\mathrm{th}}$ **then** set price $p_{\mathrm{c},i,\max}$ for queue $i$ // to control the maximum queue length
5:         **else if** $0<Q_{\mathrm{c},i}(t)<q^{\mathrm{th}}$ **then** for queue $i$, set price: // to control the average queue length

$$
\begin{cases}
\min\{p_{\mathrm{c},i}^{+/-}(k,m)+\alpha,p_{\mathrm{c},i,\max}\}, & \text{w.p. } 1/2 \\
p_{\mathrm{c},i}^{+/-}(k,m), & \text{w.p. } 1/2
\end{cases}
$$

6:         **else** set price $p_{\mathrm{c},i}^{+/-}(k,m)$ for queue $i$.
7:         **if** price$=p_{\mathrm{c},i}^{+/-}(k,m)$ **then** $n_{\mathrm{c},i}(k,m)\leftarrow n_{\mathrm{c},i}(k,m)+1$ // the number of useful samples
8:     **end for**
9:     **for** $j=1$ to $J$ **do**
10:        **if** $Q_{\mathrm{s},j}(t)\geqslant q^{\mathrm{th}}$ **then** set price $p_{\mathrm{s},j,\min}$ for queue $j$ // to control the maximum queue length
11:        **else if** $0<Q_{\mathrm{s},j}(t)<q^{\mathrm{th}}$ **then** for queue $j$, set price: // to control the average queue length

$$
\begin{cases}
\max\{p_{\mathrm{s},j}^{+/-}(k,m)-\alpha,p_{\mathrm{s},j,\min}\}, & \text{w.p. } 1/2 \\
p_{\mathrm{s},j}^{+/-}(k,m), & \text{w.p. } 1/2
\end{cases}
$$

12:        **else** set price $p_{\mathrm{s},j}^{+/-}(k,m)$ for queue $j$.
13:        **if** price$=p_{\mathrm{s},j}^{+/-}(k,m)$ **then** $n_{\mathrm{s},j}(k,m)\leftarrow n_{\mathrm{s},j}(k,m)+1$ // the number of useful samples
14:     **end for**
15:     Run the system with the above set of prices for one time slot.
16:     $t\leftarrow t+1$
17:     Terminate the algorithm when $t>T$
18: **until** for all queues $i$, $n_{\mathrm{c},i}(k,m)\geqslant N$, i.e., the price $p_{\mathrm{c},i}^{+/-}(k,m)$ is run for at least $N$ times and for all queues $j$, $n_{\mathrm{s},j}(k,m)\geqslant N$, i.e., the price $p_{\mathrm{s},j}^{+/-}(k,m)$ is run for at least $N$ times
19: **Return** $t,\big(A_{\mathrm{c},i}(t_{\mathrm{c},i}^{+/-}(k,m,n))\big)_{n\in[N],i\in\mathcal{I}},\big(A_{\mathrm{s},j}(t_{\mathrm{s},j}^{+/-}(k,m,n))\big)_{n\in[N],j\in\mathcal{J}}$

---

# H Details of the Simulation and Additional Numerical Results

## H.1 Single-Link System

We present details of the simulation and additional simulation results in a single-link system in the following.

### H.1.1 Setting

Consider a system with $I = 1$ customer queue and $J = 1$ server queue. The demand function of the customer queue is $F(\lambda) = 2(1 - \lambda)$, where $\lambda \in [0, 1]$ is the arrival rate of the customer queue. The supply function of the server queue is $G(\mu) = 2\mu$, where $\mu \in [0, 1]$ is the arrival rate of the server queue. We compare the performances among three algorithms:

- **Two-price policy** [2]: This policy assumes that *the demand and supply functions are known*. To balance queue length and regret, this policy uses the following two-price method. If the queue length is empty, it uses the prices corresponding to the optimal solution of the fluid optimization problem. If the queue is nonempty, it uses a slightly-perturbed version of the optimal prices to reduce the arrival rates.
- **Threshold policy** [1]: This policy *does not know* the demand and supply functions and uses a learning-based pricing algorithm. To balance queue length and regret, this policy uses a threshold method, rejecting arrivals when the queue length reaches or exceeds a threshold.
- **Probabilistic two-price policy** (proposed): This policy *does not know* the demand and supply functions. It uses a novel probabilistic method, which adjusts the prices to reduce the queue length with a prefixed probability when the queue is nonempty.

### H.1.2 Comparison

For the proposed algorithm, we set $\gamma = 1/6$, and $q^{\text{th}} = t^{\gamma}$, $\epsilon = t^{-2\gamma}$, $\eta = \delta = 0.2 \times t^{-\gamma}$, $\alpha = 0.2 \times t^{\gamma/2}$, $\beta = 1.0$, $e_{\text{c}} = e_{\text{s}} = 6.0 \times \max\{\delta, \eta, \epsilon\}$, $a_{\min} = 0.01$, which are of the same order as in Corollary 1. The common parameters of the three algorithm are set to be equal for fair comparison.

To compare the performance of these algorithms, we consider an objective function defined as

$$\sum_{\tau=1}^{t} \left( \sum_{i} \lambda_i(\tau) \left( F_i(\lambda_i(\tau)) - w\mathbb{E}[W_{\text{c},i}(\tau)] \right) \right.$$
$$\left. - \sum_{j} \mu_j(\tau) \left( G_j(\mu_j(\tau)) + w\mathbb{E}[W_{\text{s},j}(\tau)] \right) \right), \tag{19}$$

where $W_{\text{c},i}(\tau)$ denotes the waiting time of the customer arriving at queue $i$ at time slot $\tau$ and $W_{\text{s},j}(\tau)$ denotes the waiting time of the server arriving at queue $j$ at time slot $\tau$. The constant $w > 0$ can be viewed as the compensation the platform pays for per unit time the customer/server wait. This objective (19) is equivalent to

$$\text{Profit}(t) - w \left( \sum_{i} \sum_{k} \mathbb{E}[W_{\text{c},i,k}(t)] + \sum_{j} \sum_{k} \mathbb{E}[W_{\text{s},j,k}(t)] \right), \tag{20}$$

where $W_{\text{c},i,k}(t)$ denotes the total waiting time up to time $t$ for the $k^{\text{th}}$ customer arriving at queue $i$ , $W_{\text{s},j,k}(t)$ denotes the total waiting time up to time $t$ for the $k^{\text{th}}$ server arriving at queue $j$, and $w$ is a constant. A similar objective is also used in [2]. Note that the sum of waiting times of all customers and servers is equal to the sum of the queue lengths over time, which counts every customer and server present at each time slot. Hence, the objective (20) is equal to

$$\text{Profit}(t) - w \sum_{\tau=1}^{t} \mathbb{E} \left[ \sum_{i} Q_{\text{c},i}(\tau) + \sum_{j} Q_{\text{s},j}(\tau) \right].$$

Then, the regret of this objective is given by:

$$f(\boldsymbol{x}^*)t - \text{Profit}(t) + w \sum_{\tau=1}^{t} \mathbb{E} \left[ \sum_{i} Q_{\text{c},i}(\tau) + \sum_{j} Q_{\text{s},j}(\tau) \right]$$

$$=\mathbb{E}[R(t)] + w \sum_{\tau=1}^{t} \mathbb{E}\left[\sum_i Q_{\mathrm{c},i}(\tau) + \sum_j Q_{\mathrm{s},j}(\tau)\right]$$

$$=\mathbb{E}[R(t)] + wt\mathrm{AvgQLen}(t),$$

which is the sum of the profit regret and the holding cost.

In the simulation, we test two different $w$, $w = 0.001$ and $w = 0.01$. Note that the optimal profit at each time slot is $f(\boldsymbol{x}^*) = 0.25$ in the fluid solution. Therefore, choosing a even larger $w$ is not reasonable because further increasing $w$ will cause a negative utility even for the case where the demand and supply functions are known.

Figure 2 in Section 5 shows the performance of these algorithms and the percentage improvement of the proposed *probabilistic two-price policy* over the *threshold policy* [1] with $w = 0.001$.

The shaded areas in all figures in this paper represent $95\%$ confidence intervals, calculated from 10 independent runs using the Python function *seaborn.relplot*.

Figure 3 present the comparison results with a different weight, $w = 0.01$, on the holding cost. We observe that the proposed *probabilistic two-price policy* significantly improves upon the *threshold policy* by up to $25\%$.

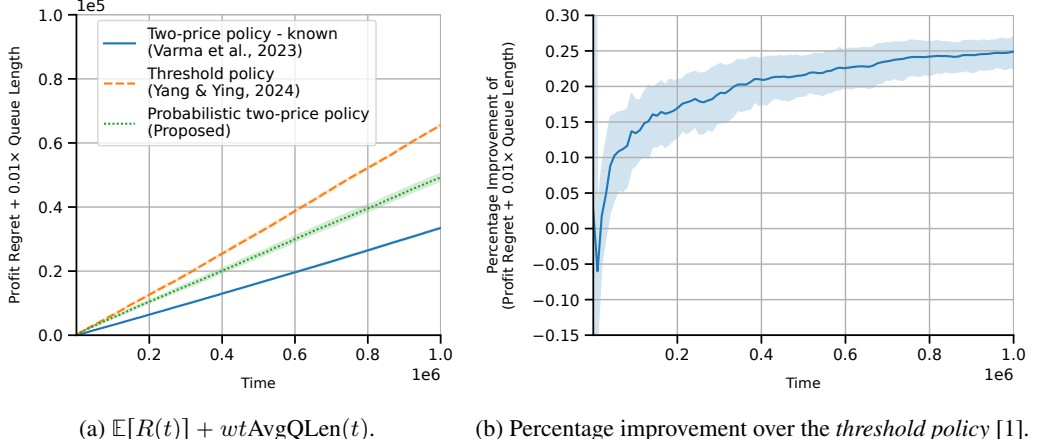

(a) $\mathbb{E}[R(t)] + wt\mathrm{AvgQLen}(t)$.       (b) Percentage improvement over the *threshold policy* [1].

Figure 3: Comparison among the *two-price policy* (no learning, known demand and supply functions) [2], the *threshold policy* [1], and the proposed *probabilistic two-price policy* with $w = 0.01$, in a single-link system.

The regret, the average queue length, and the maximum queue length of the algorithms are shown in Figure 4. Note that the X-axis and Y-axis of Figure 4 are in log-scale. The regret performance of the proposed *probabilistic two-price policy* is similar to the *threshold policy* [1]. However, we can see that the average queue length of the proposed *probabilistic two-price policy* is significantly better than the *threshold policy* because we use a novel probabilistic two-price method to reduce the average queue length without hurting the regret performance. The slope of the proposed *probabilistic two-price policy* is significantly smaller than that of the *threshold policy* [1], which is consistent with our theoretical results. As for the maximum queue length, the performance of the proposed *probabilistic two-price policy* and that of the *threshold policy* are the same, because both uses a threshold to reject the arrivals when the queue length exceeds the threshold. The *two-price policy* has a larger maximum queue length because it does not use such a hard threshold.

### H.1.3   Tradeoff Between Regret and Queue Length

Figure 5 shows the tradeoff between profit regret and average queue length for the proposed *probabilistic two-price policy* as we change the parameter $\gamma$ from $1/12$ to $1/6$. Each point in the blue curve (regret) is generated by first computing the ratio $\frac{\log_2 R(t)}{\log_2 t}$, which represents the growth order of regret with respect to time, for each time slot $t \in [10^5, 10^6]$, and then averaging over these time slots. Similarly, each point in the red curve (average queue length) is obtained by computing $\frac{\log_2 \mathrm{AvgQLen}(t)}{\log_2 t}$

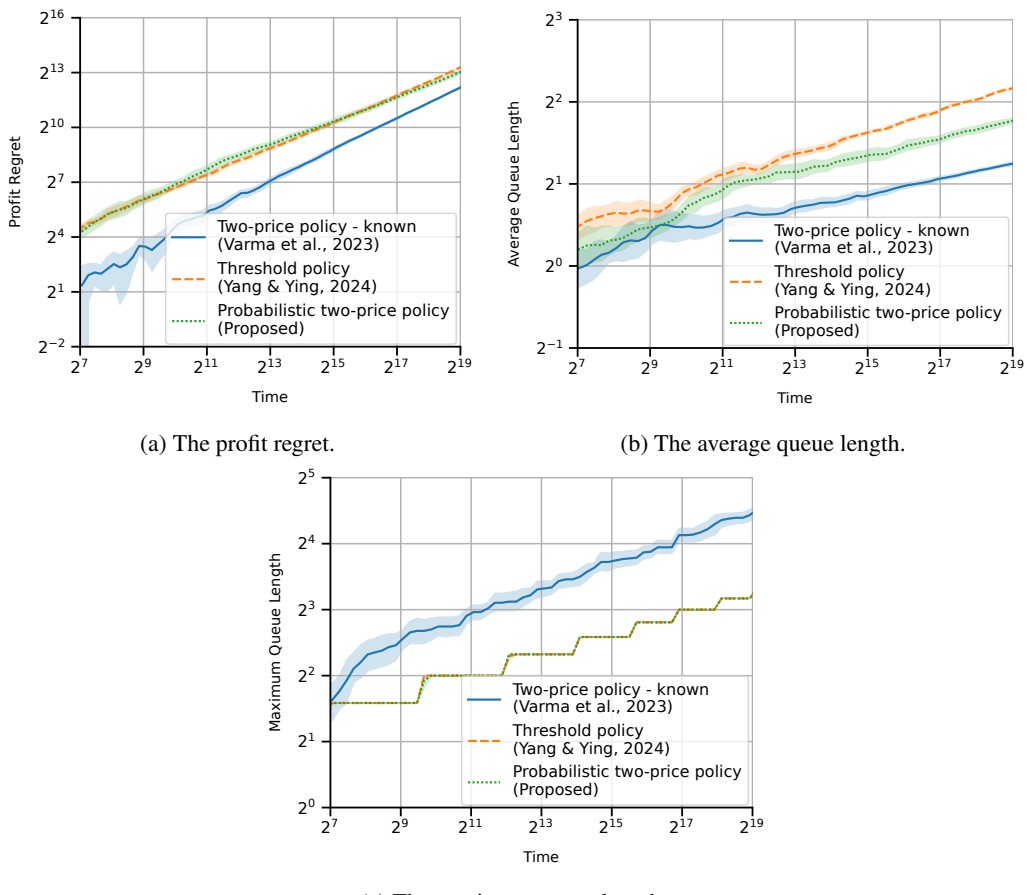

(a) The profit regret.

(b) The average queue length.

(c) The maximum queue length.

Figure 4: Comparison of regret, average queue length, and maximum queue length among the *two-price policy* (no learning, known demand and supply functions) [2], the *threshold policy* [1], and the proposed *probabilistic two-price policy*, in a single-link system.

and averaging over the same time slots. We observe that as $\gamma$ increases, the growth order of regret decreases while the growth order of average queue length increases, confirming the tradeoff predicted by our theoretical results. By a simple least square fitting, we can obtain the slopes of the curves in Figure 5. The fitted function of the blue curve (regret) is $y = -1.484x + 0.927$ and the fitted function of the red curve (average queue length) is $y = 0.615x - 0.011$, which implies that the regret is $T^{0.927-1.484\gamma}$ and the average queue length is $T^{0.615\gamma-0.011}$. Although they are not exactly equal to $T^{1-\gamma}$ and $T^{0.5\gamma}$, $-1.484/0.615 \approx -2.4$, which is similar to $-1/0.5 = -2$. This result confirms the near-optimal tradeoff, $\tilde{O}(T^{1-\gamma})$ versus $\tilde{O}(T^{\gamma/2})$, in Corollary 1.

### H.1.4 Sensitivity to Parameter Tuning

We test the sensitivity of the performance of the proposed *probabilistic two-price policy* to parameter tuning. We test the multiplicative constant of the parameters $\epsilon$, $\delta$, and $\eta$. Figure 6 shows the performances for different multiplicative constants in the parameter $\epsilon$, i.e., the constant $C$ for $\epsilon = Ct^{-2\gamma}$. We can see that the performances are almost the same. Figure 7 shows the performances for different multiplicative constants in the parameter $\delta$, i.e., the constant $C$ for $\delta = Ct^{-\gamma}$. Figure 8 shows the performances for different multiplicative constants in the parameter $\eta$, i.e., the constant $C$ for $\eta = Ct^{-\gamma}$.

### H.2 Multi-Link System

We present simulation results in a multi-link system in the following.

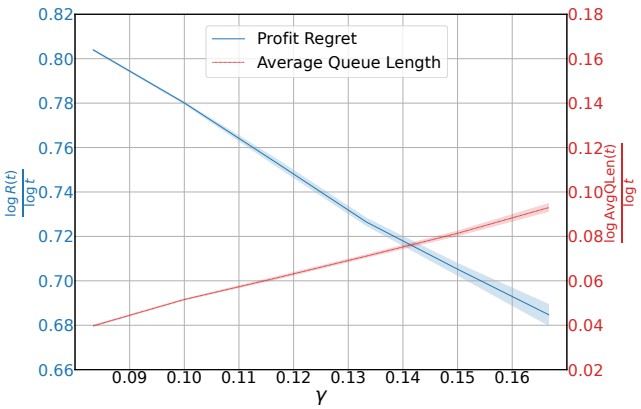

Figure 5: Tradeoff between regret and average queue length.

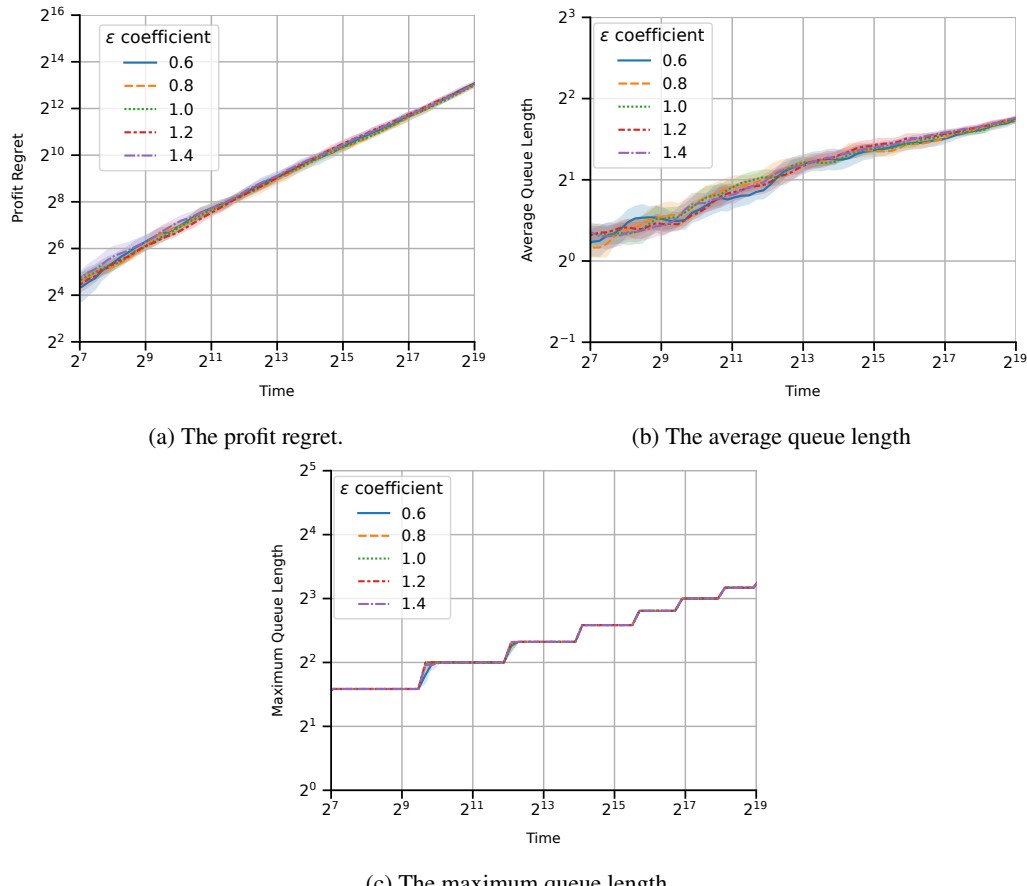

(a) The profit regret.

(b) The average queue length

(c) The maximum queue length

Figure 6: Sensitivity testing for different $\epsilon$ for the proposed *probabilistic two-price policy*.

### H.2.1 Setting

Consider a system with three types of customers and three types of servers ($I = 3, J = 3$). The compatibility graph $\mathcal{E}$ is $\mathcal{E} = \{(1,1), (1,2), (1,3), (2,1), (2,2), (3,2), (3,3)\}$. The demand functions are $F_i(\lambda_i) = 2(1 - \lambda_i)$ for all $i = 1, 2, 3$. The supply functions are $G_j(\mu_j) = 2\mu_j$ for all $j = 1, 2, 3$. We compare the same three algorithms as in the single-link simulation, i.e., the *two-price policy* [2], the *threshold policy* [1], and the proposed *probabilistic two-price policy*.

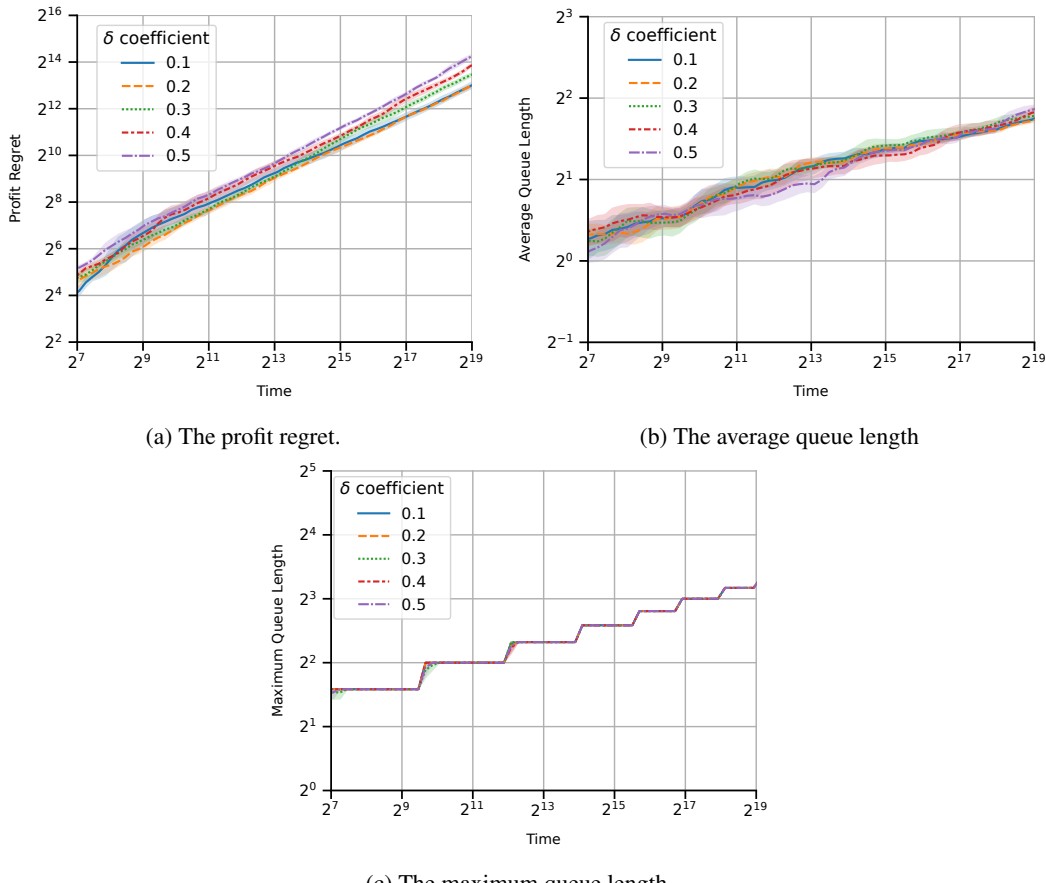

(a) The profit regret.

(b) The average queue length

(c) The maximum queue length

Figure 7: Sensitivity testing for different $\delta$ for the proposed *probabilistic two-price policy*.

### H.2.2    Comparison

For the proposed algorithm, we set $\gamma = 1/6$, and $q^{\text{th}} = t^\gamma$, $\epsilon = t^{-2\gamma}$, $\delta = 0.2 \times t^{-\gamma}$, $\eta = 0.1 \times t^{-\gamma}$, $\alpha = 0.2 \times t^{\gamma/2}$, $\beta = 1.0$, $e_c = e_s = 8.0 \times \max\{\delta, \eta, \epsilon\}$, $a_{\min} = 0.01$, which are of the same order as in Corollary 1. The common parameters of the three algorithm are set to be equal for fair comparison.

Consider the same objective as that in the single-link simulation.

Figure 9 present the comparison results with $w = 0.01$, on the holding cost. We observe that the proposed *probabilistic two-price policy* significantly improves upon the *threshold policy* [1] by up to 34% for large $t$. Figure 10 present the comparison results with a different weight, $w = 0.005$. We observe that the proposed *probabilistic two-price policy* significantly improves upon the *threshold policy* by up to 12% for large $t$. We observe that in the multi-link system, initially for small $t$, the proposed *probabilistic two-price policy* may not perform as good as the *threshold policy*. As $t$ increases, our proposed policy quickly outperforms the *threshold policy*. This is because, as $t$ increases, the improvement in queue length performance under our proposed algorithm becomes more significant. Figure 11 shows the performance of these algorithms with a much smaller weight, $w = 0.001$. In this case, although our proposed policy does not perform as good as the *threshold policy*, it improves as $t$ increases. The comparison depends on the weight $w$ on the holding cost and the time horizon.

The regret, the average queue length, and the maximum queue length of the algorithms are shown in Figure 12. The observations are similar to the single-link simulation. The regret performance of the proposed *probabilistic two-price policy* is similar to the *threshold policy* [1]. However, the average queue length of the proposed *probabilistic two-price policy* is significantly better than the *threshold policy*. As for the maximum queue length, the performance of the proposed *probabilistic two-price*

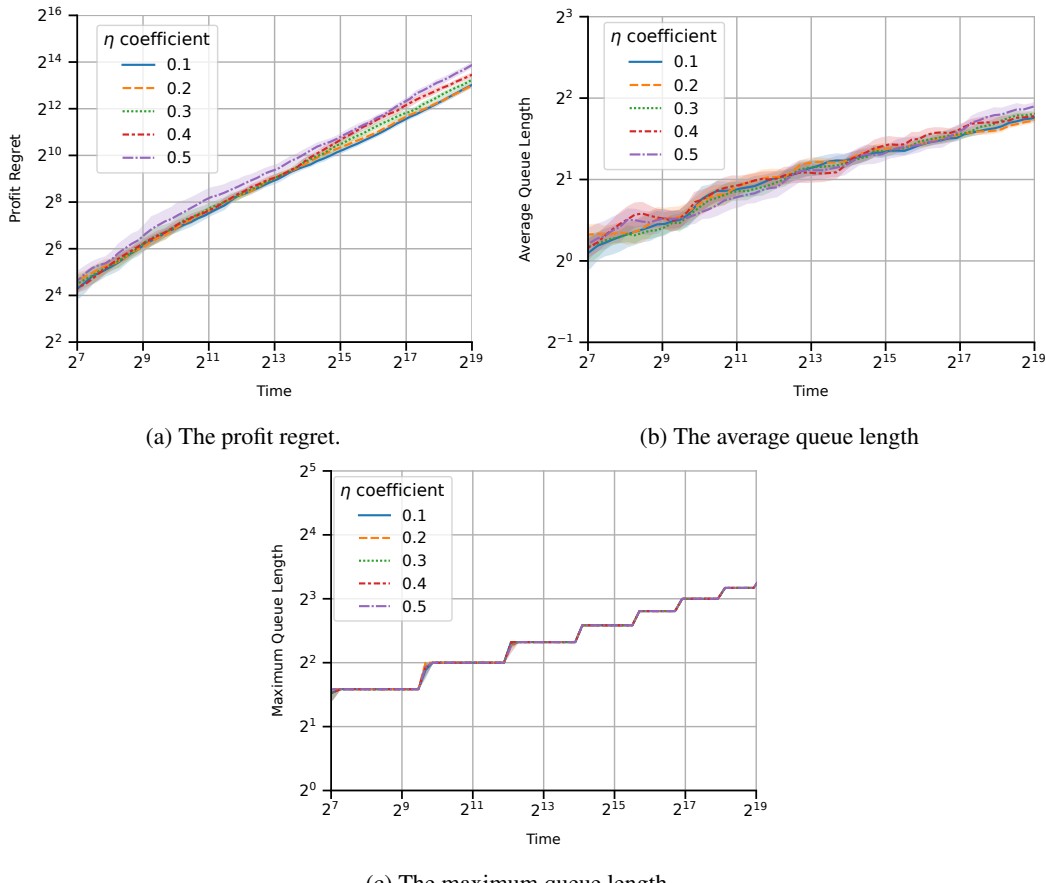

(a) The profit regret.

(b) The average queue length

(c) The maximum queue length

Figure 8: Sensitivity testing for different $\eta$ for the proposed *probabilistic two-price policy*.

*policy* and that of the *threshold policy* are the same. The *two-price policy* has a larger maximum queue length.

### H.3   Compute Resources

All the simulations are not compute-intensive, and all results can be reproduced on a standard personal computer. On a personal computer with Intel Core i5-9400 CPU @2.90GHz, 10 runs of the simulation of the single-link system ($T = 10^6$) for all three algorithms take approximately 20 minutes, and 10 runs of the simulation of the multi-link system ($T = 10^7$) for all three algorithms take approximately 6 hours.

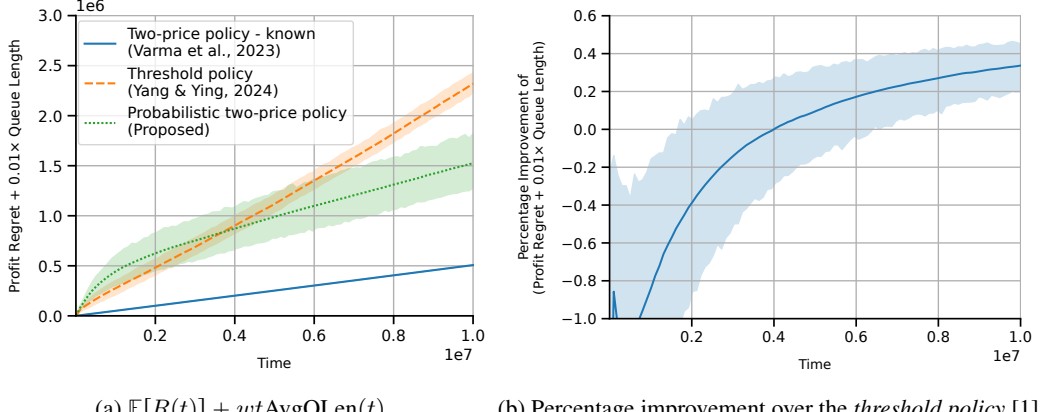

(a) $\mathbb{E}[R(t)] + wt\mathrm{AvgQLen}(t)$.

(b) Percentage improvement over the *threshold policy* [1].

Figure 9: Comparison among the *two-price policy* (no learning, known demand and supply functions) [2], the *threshold policy* [1], and the proposed *probabilistic two-price policy* with $w = 0.01$, in a multi-link system.

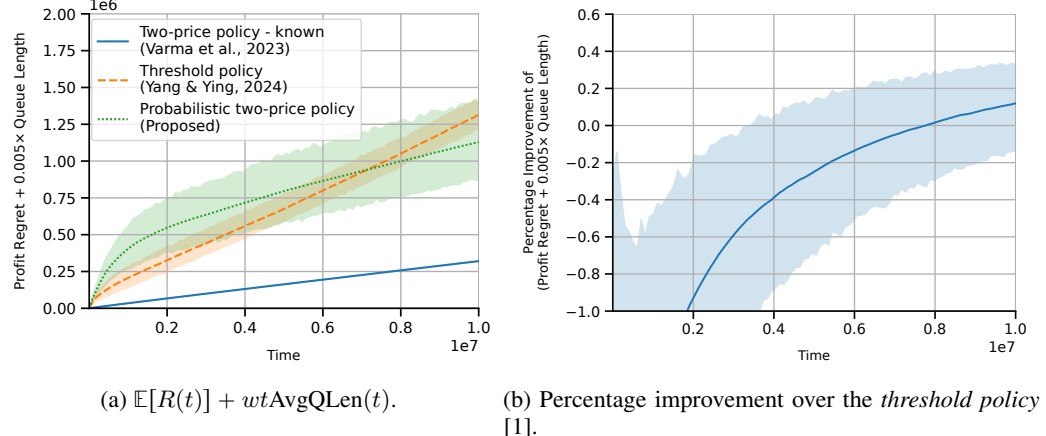

(a) $\mathbb{E}[R(t)] + wt\mathrm{AvgQLen}(t)$.

(b) Percentage improvement over the *threshold policy* [1].

Figure 10: Comparison among the *two-price policy* (no learning, known demand and supply functions) [2], the *threshold policy* [1], and the proposed *probabilistic two-price policy* with $w = 0.005$, in a multi-link system.

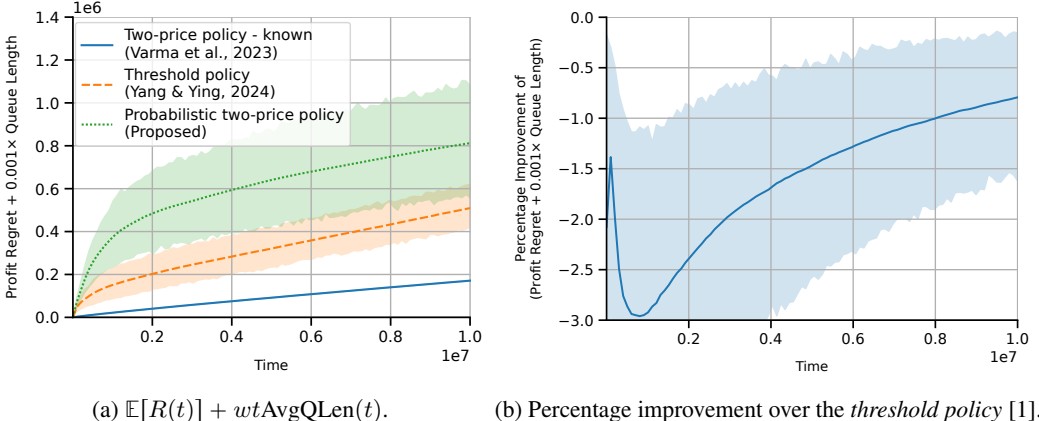

(a) $\mathbb{E}[R(t)] + wt\mathrm{AvgQLen}(t)$.

(b) Percentage improvement over the *threshold policy* [1].

Figure 11: Comparison among the *two-price policy* (no learning, known demand and supply functions) [2], the *threshold policy* [1], and the proposed *probabilistic two-price policy* with $w = 0.001$, in a multi-link system.

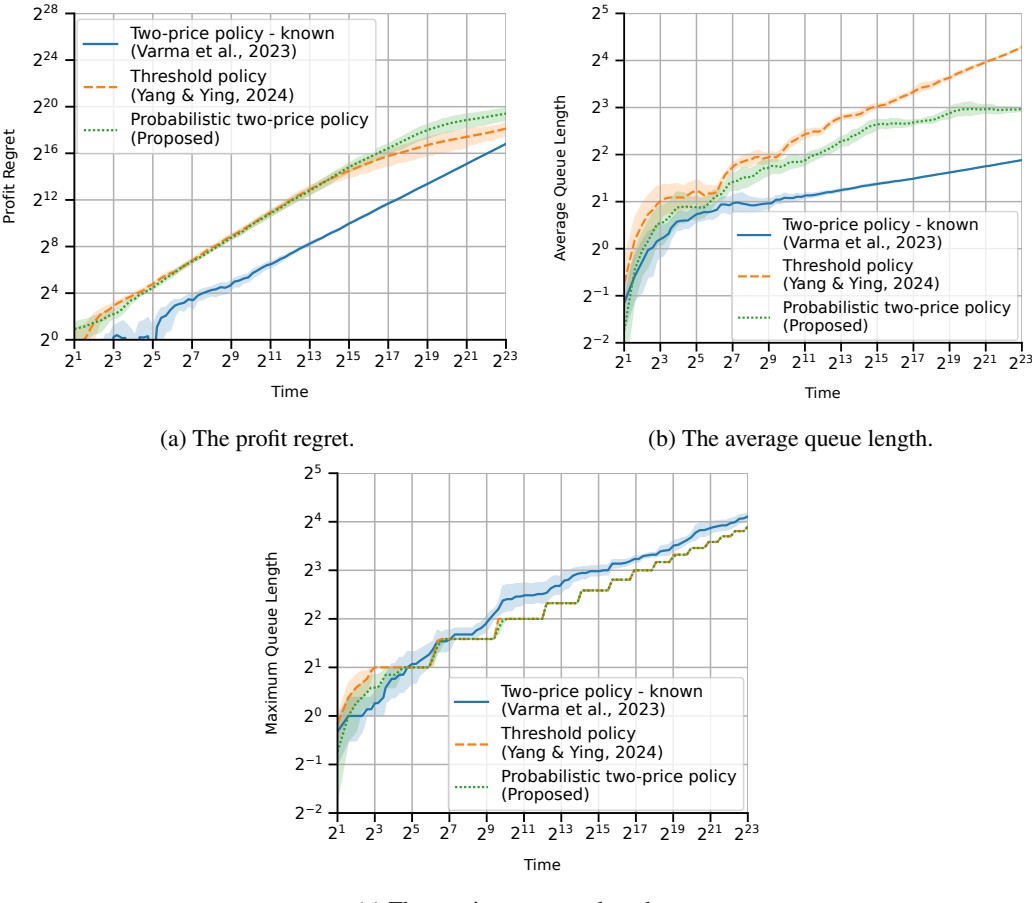

(a) The profit regret.

(b) The average queue length.

(c) The maximum queue length.

Figure 12: Comparison of regret, average queue length, and maximum queue length among the *two-price policy* (no learning, known demand and supply functions) [2], the *threshold policy* [1], and the proposed *probabilistic two-price policy*, in a multi-link system.

# I  Societal Impacts

This paper presents work whose goal is to advance the fields of online learning in queueing systems. There are potential positive societal impacts, including improving efficiency and promoting profitability in applications like ride-hailing, meal-delivery, and crowdsourcing. We are not aware of any negative societal impacts of this work.

