# OpenReview forum: "Near-Optimal Regret-Queue Length Tradeoff in Online Learning for Two-Sided Markets"
_NeurIPS.cc/2025/Conference — NeurIPS 2025 poster_

### Official Review · Reviewer_QxRW · 2025-07-01

**Clarity:** 2
**Significance:** 4
**Originality:** 3
**Rating:** 5
**Confidence:** 3

**Summary:**

The paper studies a queuing system where customers and servers have probabilities to join the system at each time step. These probabilities depend on prices set by the learner, the higher the price of the customer, the smaller probability the customer joins, and conversely for the server. The mapping between prices and probabilities of arrival is unknown, contrary to earlier references.

The goal of the learner is to provide a Pareto-optimal tradeoff between the two following objectives. (1) maximizing the profit, which is computed from the fact that the learner pays the servers (with the prices set to the servers) and earns money from the customers (again with the prices set to the customers). (2) minimising the queue length, that is taking care that not too many customers and servers join the system and do not exit it quickly.

The paper defines a notion of cumulated regret, as the difference between the achieved profit and a profit obtained by optimising a "static" version of the queuing system, where the optimisation necessitates to know the mapping between prices and probabilities of arrival. The paper proposes a new algorithm that has an hyper parameter. Varying the hyper-parameter enables to explore the Pareto front of the problem, by giving Pareto-optimal pairs of regret / queue length. The algorithm has a learning component of the mapping between prices and probabilities of arrival. The paper provides a family of lower bounds that confirm Pareto-optimality. Finally, numerical experiments are provided, confirming the improvement brought by the new algorithm, over existing ones.

**Questions:**

I am not sure to understand well the sentence on Lines 79-80.

Some readers may not be familiar with the definition of a M/G/1 queue.

Line 93 sets(sides), a space is missing.

Line 100, this is for all i,j?

Line 110-117. I am troubled by the mapping from arrival rate to price. This can erroneously make the reader think that prices are caused by arrival rates? However, I understand that the learner chooses the prices freely, and that the chosen price causes an arrival rate. Would it be clearer to rather introduce mappings from prices to arrival rate?

Lemma 1 is for I = J = 1. Could there be a similar lower bound for general values of I and J?

In Figure 2 which value of the hyperparameter gamma was used?

**Ethical Concerns:**

["NO or VERY MINOR ethics concerns only"]

**Final Justification:**

Given the author's response to me, and to the other reviews, I am quite confident in my positive evaluation of the paper, in the end.

**Limitations:**

YES

**Paper Formatting Concerns:**

No concerns.

**Quality:**

3

**Strengths And Weaknesses:**

STRENGTHS I think the topic is relevant, interesting, and elegant, being at the crossroads between learning and queuing theory. The theoretical setting is precisely introduced and the proofs are significant and appear to be challenging. The results provide an improvement on the literature, as the previous literature either provided guarantees that were not Pareto-optimal, or worked with known mappings between prices and probabilities of arrival.


WEAKNESSES This is just a minor weakness in my opinion: the paper uses a significant space of the main text to explain the challenges to establish their results (end of Section 2, end of Section 4). This space could have been used to provide more technical details or pedagogical explanations (like the very clear Figure 1).

---

> ### Author Rebuttal · Authors · 2025-07-30
>
> We thank the reviewer for the positive comments and valuable suggestions.
>
> **Comment:** "I am not sure to understand well the sentence on Lines 79-80."
>
> **Response:** Thank you for catching the grammatical error and for letting us know that the original wording was hard to follow.
> We will rewrite the sentence as follows:
>
> "One line of work views the queueing system as a Markov decision process (MDP) and uses reinforcement learning methods in an attempt to learn a global optimal policy [15, 16]. These methods are broadly applicable to many types of queueing systems."
>
> **Comment:**
> "Some readers may not be familiar with the definition of a M/G/1 queue."
>
> **Response:**
> We will rewrite the sentence as follows:
>
> "Closest to ours is [19] that learns to price for a queueing system with a single server and a single queue, where arrivals follow a Poisson process and service times are independent and identically distributed with a general distribution."
>
> **Comment:** "Line 93 sets(sides), a space is missing."
>
> **Response:** Thanks for pointing out. We will add a space in the revision.
>
> **Comment:** "Line 100, this is for all i,j?"
>
> **Response:** Yes, we observe queue lengths for all $i,j$. We will clarify it in the revision.
>
> **Comment:** "Line 110-117. I am troubled by the mapping from arrival rate to price. This can erroneously make the reader think that prices are caused by arrival rates? However, I understand that the learner chooses the prices freely, and that the chosen price causes an arrival rate. Would it be clearer to rather introduce mappings from prices to arrival rate?"
>
> **Response:** Thanks for pointing out. We originally wanted to keep the notation consistent with [1,2]. In the revision, we will first introduce mappings from prices to arrival rate and then introduce the inverse mapping.
>
> **Comment:** "Lemma 1 is for I = J = 1. Could there be a similar lower bound for general values of I and J?"
>
> **Response:** Thank you for your comment. While our lower bound is restricted to the special case of $I=J=1$, it is sufficient to conclude the worst-case near-optimality of our upper bound. While we do not prove a formal lower bound for general $I, J$, intuitively, such a lower bound should hold generally. In particular, a general two-sided queueing model with a general compatibility graph should be worse off compared to the two-sided queueing model on a complete bipartite graph $G(I \cup J, I \times J)$. In addition, this complete bipartite graph model behaves similarly to a two-sided queueing model with $I=J=1$, where all the waiting customers and waiting servers are clubbed into a single queue [resource pooling]. Thus, the regret and queue lengths for a general two-sided queueing model could be lower bounded by its corresponding ``resource pooled'' single-link ($I=J=1$) two-sided queue.
>
> **Comment:** "In Figure 2 which value of the hyperparameter gamma was used?"
>
> **Response:** We set $\gamma=1/6$ in Figure 2, which is mentioned in Appendix M.1.2. We will add this in the main text in the revision.

---

> > ### Comment · Reviewer_QxRW · 2025-08-01
> > **Response acknowledgement**
> >
> > I thank the authors for their response. I appreciate the clarifications.
> > I am happy to maintain my positive evaluation.

---

> > > ### Author Response · Authors · 2025-08-05
> > > **Thanks!**
> > >
> > > Thanks again for your great questions and positive comments!

---

### Official Review · Reviewer_jgX5 · 2025-07-02

**Clarity:** 4
**Significance:** 3
**Originality:** 3
**Rating:** 5
**Confidence:** 4

**Summary:**

This paper studies pricing and matching in a two-sided market with both demand queues and server queues. The authors propose an online algorithm and analyze the corresponding performance metrics, including upper bounds on regret, average queue length, and maximum queue length. Since these metrics inherently conflict with one another, the authors also characterize the trade-off between regret and average queue length and show that the resulting trade-off is optimal.

**Questions:**

The authors compare their results to Varma et al. (2023) and Yang & Ying (2024) throughout the paper. From the perspective of algorithm design, how does the proposed algorithm fundamentally differ from the one in Yang & Ying (2024)?

Echoing the weakness above, I would appreciate a more concrete definition or intuitive explanation of the Pareto frontier. The authors repeatedly emphasize that $(1 - \gamma)$ and $\gamma$ is not an optimal trade-off, while $(1 - \gamma)$ and $\gamma/2$ is optimal. Could the authors clarify what this trade-off represents and why one is considered better than the other?

My next question concerns the definition of profit at the bottom of Page 3. According to the current definition, any customer who joins the queue contributes to profit, regardless of when (or whether) the customer is served or how long the wait time is. Suppose the algorithm sets a very low price in the last period $T$, causing many customers to join the queue. Under the regret definition in Equation (5), would this result in negative regret? Could the authors clarify how this edge case is handled?

In Lines 201 and 202, why does the equation $Q_c(t)^2 + Q_s(t)^2 = (Q_c(t) + Q_s(t))^2$ hold? What is the intuition behind this equality?

**Ethical Concerns:**

["NO or VERY MINOR ethics concerns only"]

**Final Justification:**

I appreciate the authors’ detailed response and their efforts to clarify key aspects of the paper. My questions are all well addressed, particularly regarding the algorithmic contribution. Overall, I find this paper will be of interest to the community, and thus I would like to keep my original positive rating.

**Limitations:**

Yes.

**Paper Formatting Concerns:**

Na.

**Quality:**

4

**Strengths And Weaknesses:**

Strengths:

The paper is well written, and the motivation, problem formulation, assumptions, and algorithm analysis are all clearly presented. In particular, the authors provide thoughtful and meaningful discussions to justify their assumptions and to contextualize the results in comparison with existing literature.

Weaknesses:

The performance metric, i.e., the Pareto frontier, is not explicitly defined in the paper. I suggest that the authors introduce this concept, even briefly, to help readers who are unfamiliar with multi-objective trade-off analysis. For example, the authors repeatedly state that the trade-off between $(1 - \gamma)$ and $\gamma$ is suboptimal, while $(1 - \gamma)$ and $\gamma/2$ is optimal. Without a formal definition or intuitive explanation of the Pareto frontier, this claim is not clear to readers with limited background in this area.

---

> ### Author Rebuttal · Authors · 2025-07-30
>
> We thank the reviewer for the positive comments and valuable suggestions.
>
> **Comment:** "From the perspective of algorithm design, how does the proposed algorithm fundamentally differ from the one in Yang \& Ying (2024)?"
>
> **Response:** The probabilistic two-price policy is the fundamental difference compared to (Yang \& Ying, 2024).
> Yang \& Ying (2024) employs a pricing policy with an admission control threshold. When the queue length reaches or exceeds the threshold, any new arrivals will be rejected to control the queue length.
> This approach fails to achieve the optimal $1-\gamma$ versus $\gamma/2$ tradeoff as discussed in on Page 6, Lines 215–220.
> The two-price policy in (Varma et al., 2023)---without our probabilistic component---is insufficient in the learning setting.
> In (Varma et al., 2023), the arrival rate is reduced by a small amount when the queue exceeds a threshold, inducing a negative drift that reduces the queue lengths in expectation. However, in the learning setting, price adjustments not only control queues but also affect the learning process. Adjusting prices introduces bias into arrival samples, degrading gradient estimates and increasing regret.
> Our novel probabilistic two-price policy addresses this by reducing arrival rates by $\Theta(\alpha)$ (via adjusting prices) with a constant probability when the queue is nonempty. This probabilistic component ensures both negative drift and fast learning with unbiased samples. By tuning $\alpha$, we achieve a near-optimal balance between queue length and regret. (See Page 8, Lines 336–352.)
>
> **Comment:** "Echoing the weakness above, I would appreciate a more concrete definition or intuitive explanation of the Pareto frontier. The authors repeatedly emphasize that $1-\gamma$ and $\gamma$ is not an optimal trade-off, while $1-\gamma$ and $\gamma/2$ is optimal. Could the authors clarify what this trade-off represents and why one is considered better than the other?"
>
> **Response:** We will add the following definition in the revision.
> Let $E_{\pi,d}[R(T)]$ denote the expected profit regret under policy $\pi$ and problem instance $d$. The problem instance $d$ specifies the number of customer queues, the number of server queues, the set of compatible links, and the demand and supply functions. Let $\mathrm{AvgQLen}_{\pi,d}(T)$ denote the average queue length under policy $\pi$ and problem instance $d$.
> Let $(x(\pi),y(\pi))$ denote the exponents of the objective values under the worst case, i.e.,
>
> $$
> x(\pi)\coloneqq \limsup_{T\rightarrow\infty} \log_T (\sup_d \mathbb{E}_{\pi,d}[R(T)])
> $$
>
> and
> $$
> y(\pi)\coloneqq \limsup_{T\rightarrow\infty} \log_T (\sup_d \mathrm{AvgQLen}_{\pi,d}(T)).
> $$
>
> A point $(x(\pi_1),y(\pi_1))$ is said to be better than another point $(x(\pi_2),y(\pi_2))$ if $x(\pi_1) \le x(\pi_2)$ and $y(\pi_1)\le y(\pi_2)$
> with at least one inequality strict.
>
> A point $(x(\pi),y(\pi))$ is Pareto-optimal if there is no other policy $\pi'$ that satisfies $x(\pi') \le x(\pi)$ and $y(\pi')\le y(\pi)$
> with at least one inequality strict. The set of all Pareto-optimal points forms the Pareto frontier.
>
> According to the above definition, $(1-\gamma,\gamma/2)$ is better than $(1-\gamma,\gamma)$ since $\gamma/2 < \gamma$, which means that we can achieve orderwise smaller average queue length while maintaining the same order of profit regret. The $1-\gamma$ versus $\gamma/2$ tradeoff means that as we increase the value of $\gamma$, the profit regret $\tilde{\Theta}(T^{1-\gamma})$ decreases while the average queue length $\tilde{\Theta}(T^{\gamma/2})$ increases. Lemma 1 shows that this tradeoff is Pareto-optimal for $\gamma\in[0, 1/2]$ for a large class of policies.
>
> **Comment:** "According to the current definition, any customer who joins the queue contributes to profit, regardless of when (or whether) the customer is served or how long the wait time is. Suppose the algorithm sets a very low price in the last period $T$, causing many customers to join the queue. Under the regret definition in Equation (5), would this result in negative regret? Could the authors clarify how this edge case is handled?"
>
> **Response:** Since our model is discrete-time and the arrival in each time slot/period is Bernoulli, at most one customer will arrive for each queue in each time slot. Hence, only changing the policy in the last period will not significantly influence the regret, and the influence is negligible when $T$ is large.
> If we set this price for many periods to reduce the profit regret, the queue length will increase significantly.
> The profit regret defined in (5) could possibly be negative under certain extreme policies, but this comes at the cost of very large queue lengths, which are impractical in real systems. Lemma 1 shows that the regret will not be negative if the average queue length is small.
>
> **Comment:** "In Lines 201 and 202, why does the equation $Q_c(t)^2 + Q_s(t)^2 = (Q_c(t) + Q_s(t))^2$ hold? What is the intuition behind this equality?"
>
> **Response:** This equation holds since $Q_c(t)Q_s(t)=0$ for any $t$.
> The intuition is that if both queues are nonempty, the customers and servers should have already been matched under the greedy matching policy, so such a situation should not occur.

---

> > ### Comment · Reviewer_jgX5 · 2025-08-01
> >
> > I appreciate the authors’ response and the clarifications provided. I have no further questions and will maintain my current rating.

---

> > > ### Author Response · Authors · 2025-08-05
> > > **Thanks!**
> > >
> > > Thanks again for your great questions and positive comments!

---

### Official Review · Reviewer_6iW3 · 2025-07-03

**Clarity:** 3
**Significance:** 2
**Originality:** 2
**Rating:** 4
**Confidence:** 4

**Summary:**

This paper studies the two-sided matching model developed in [1], where multi-class arrivals occur on each side (think riders on one side and drivers on the other). The platform controls "pricing" (affects arrival rates) and "matching" decisions in each round (the latter is subject to a compatibility graph) to ensure long-term stability of the system; matched units exit the system immediately. Metrics of interest are (i) Profit over T rounds; (ii) Avg. Queue Length over T rounds; and (iii) Max Queue Length over T rounds, starting from an empty system at t=0. The authors establish a near-optimal trade-off between profit-regret and avg. queue length, which is achieved by their pricing/matching algorithm. A bound on the max queue length is also provided. Numerical experiments are provided showing effectiveness of the proposed solution vis-a-vis the baseline policies in [1] and [2].

**Questions:**

Please see the previous section.

**Ethical Concerns:**

["NO or VERY MINOR ethics concerns only"]

**Final Justification:**

The authors provided some additional context which was useful. There are technical merits to the algorithm/analysis that warrant an accept. That said, there is room to improve the writing/positioning for a broader audience especially given that this paper builds upon prior work [1,2] that itself is somewhat niche.

**Limitations:**

Yes.

**Paper Formatting Concerns:**

None.

**Quality:**

2

**Strengths And Weaknesses:**

Strengths:
1. Paper is well-written and accessible.
2. The proposed pricing/matching algorithm achieves a near-optimal trade-off between profit-regret and avg. queue length, while matching the bound on max queue length from prior work.

Weaknesses:
1. The model and matching algorithm are identical to [1], lacking any novelty. The innovation lies in the proposed pricing algorithm, which is a probabilistic adaption of the two price policy in [2] (for known supply and demand) to the online learning setting -- this adaption achieves the desired trade-off.
2. A very similar (near-optimal) trade-off is already established in [2], albeit under a slightly different (but related) scaling parameter.
3. It appears that the paper borrows heavily from the modeling, algorithmic, and analytic machinery already built in [1,2].
4. The underlying model itself (in my opinion) is not general enough (like K-MAB for instance) to warrant an independent study on a question already addressed in [1] to a large extent.

Of course, the paper identifies a gap (namely, the trade-off implied by results in [1] being weaker than that established in [2]) and addresses it via a neat solution. But weaknesses pointed out above give it an incremental look that may not interest many.

That said, I'm open to revising my scores pending a convincing author rebuttal.

---

> ### Author Rebuttal · Authors · 2025-07-30
>
> We greatly appreciate the reviewer's openness in reevaluating the score. We first would like to comment that it is indeed true that both the model and the matching algorithm are identical to [1]. In fact, both the model and MaxWeight matching are standard in the two-sided queue literature, not just in this paper and [1,2]. The contribution of this paper is on learning and pricing, not on modeling and matching. Our new probabilistic two-price policy allows us to achieve the near-optimal regret-queue tradeoff and close a significant gap in the literature. The proof machinery follows Lyapunov drift analysis, which is standard in stochastic control and online learning. However, while the structure of the proof and the machinery may look standard and similar to [1], like using the Lyapunov analysis for stability in control, every proof requires a customized analysis of the drift to obtain the desired bound. The drift analysis is not a plug-and-play.  Proving our result requires several technical innovations that will be explained later in this rebuttal.
>
> **Comment:** "A very similar (near-optimal) trade-off is already established in [2], albeit under a slightly different (but related) scaling parameter."
>
> **Response:** Thank you for your comment. As you rightly pointed out, the near-optimal trade-off scaling in the case of known demand and supply curves is known in the literature [2]. However, prior to our work, it was unclear whether the same trade-off continues to hold in the learning setting when these curves are unknown. We fill this gap by showing that a similar trade-off continues to hold, albeit the range of $\gamma$ is restricted to $[0, 1/6]$, which is primarily because of the errors introduced due to learning. To establish this result, we needed to overcome several technical challenges as compared to [2], and we outline the two main challenges below:
>
> - As the demand and supply curves are known in [2] and as they define a one-to-one mapping between the prices and the arrival rates, the analysis in [2] directly works with the arrival rates without loss of generality. However, in the learning setting, we are restricted to controlling the prices and observing the arrival rates as a response. This restriction introduces the first major challenge: the optimization problem (1)-(3) is nonconvex in the space of prices, and the constraints are nonlinear in terms of the prices, and it involves the unknown demand and supply curves. Thus, we cannot directly implement a zero-order learning scheme on (1)-(3), and so, we are forced to go back and forth in terms of prices and (estimated) arrival rates.
> - The objective of [2] is to understand the regret-queue-length trade-off in the steady state. While in the learning setting, it doesn't make sense to consider such a long-term objective to analyze the cost of learning. Thus, we carry out a finite-time analysis of the underlying queueing system, which leads to several technical challenges. Please refer to the comment below for a more detailed discussion on the technical challenges.
>
>
> **Comment:** "It appears that the paper borrows heavily from the modeling, algorithmic, and analytic machinery already built in [1,2]."
>
> **Response:** The novel algorithmic and analytic contributions compared to [1,2] include:
>
> - **Algorithmic contributions:** Our novel probabilistic two-price policy is fundamental to achieving the near-optimal tradeoff between profit regret and average queue length. In particular, the pricing policies in [1] and [2] are unable to achieve this trade-off in the learning setting. Specifically, [1] introduces a static pricing policy with a limited waiting area, which achieves a sub-optimal trade-off scaling as it lacks a dynamic component in the pricing policy. We have discussed this briefly on Page 6, Lines 215–220 in the paper. On the other hand, while the two-price policy of [2] is dynamic, it still may fail to achieve the trade-off in the learning setting. In the learning setting, price adjustments not only control queues but also affect the learning process. Adjusting prices introduces bias into arrival samples, degrading gradient estimates and increasing regret. Our probabilistic policy addresses this by reducing arrival rates by $\Theta(\alpha)$ (via adjusting prices) with a constant probability when the queue is nonempty. This probabilistic component ensures both negative drift and fast learning with unbiased samples. By tuning $\alpha$, we achieve a near-optimal balance between queue length and regret. (See Page 8, Lines 336–352.) We continue to use the MaxWeight matching algorithm as in [1, 2], as it is known to perform well and is agnostic to the demand and supply curves; thus, it can be easily implemented in the learning setting. Note that MaxWeight is a standard approach that has been used beyond two-sided queues, including high-speed switches, wireless scheduling, etc.
> - **Technical/Analytic contributions:** Proving that the probabilistic two-price policy achieves a near-optimal tradeoff requires addressing new analytical challenges beyond [1,2], as outlined on Pages 8–9, Lines 353–364. In particular, our approach has the following two technical innovations that go beyond both [1] and [2], which are only possible due to our probabilistic two-price policy:
>     - First, we bound the number of biased samples caused by price adjustments. This is handled by showing, via Wald’s lemma (Appendix I.4.1), that the time spent on such adjustments is bounded while the time spent in collecting unbiased samples is sufficient.
>     - Second, we bound the regret from these adjustments by analyzing both first-order and second-order errors. Our bounds on the first-order errors are much sharper compared to [1], resulting in a better regret-queue-length trade-off. Moreover, unlike [2], this requires a more involved analysis in our finite-time learning setting, where the actual arrival rates are suboptimal, vary across iterations, and only approximately satisfy the balance equation. We address this by combining the KKT conditions (Appendix I.3) and a drift argument (Lemma 5, proof in Appendix L).
> - **Fundamental tradeoff:** Lemma 1 establishes a new lower bound beyond [1,2], identifying the fundamental tradeoff between regret and queue length in the finite-time setting.
>
>
>
> **Comment:** "The underlying model itself (in my opinion) is not general enough (like K-MAB for instance) to warrant an independent study on a question already addressed in [1] to a large extent."
>
> **Response:** Thank you for your comment. We believe that the two-sided queueing model with pricing and matching applies to many real-world systems, such as ride-hailing, meal delivery, and crowdsourcing platforms.
> For example, in ride-hailing, riders and drivers form two sides of queues awaiting matching. The platform charges riders and pays drivers upon arrival, with prices influencing their arrival rates. A similar structure holds for customers and couriers in meal delivery, and for tasks and workers in crowdsourcing. In these systems, the demand and supply curves are often unknown, and one needs to constantly learn them using the historical data. One approach is then to use the estimated curves to solve the optimal pricing scheme (see comment \#3 of Reviewer C861 for more discussion), but this approach may lead to linearly growing regret and queue length in the estimation process. Such a difficulty motivated the analysis in this work, which combines estimation with optimization.
>
> Also note that our probabilistic approach decouples the complex correlations between *fast learning* (obtaining useful samples for learning) and *low queueing delay*, while providing the flexibility to tune the policy parameters to optimize objectives such as *profit maximization*. We believe such an idea is quite general and may be broadly applicable to online learning in queueing systems. We briefly discuss the potential of our approach on Page 14, Lines 537-546.

---

> > ### Comment · Reviewer_6iW3 · 2025-08-01
> >
> > I thank the authors for their comprehensive explanation and providing some additional context which was useful. I'm happy to revise my score upward and vote for acceptance. But I would still insist that the authors revise the paper to reflect the essence of this rebuttal more clearly. The current version risks leaving an incremental impression outside of a niche audience.

---

> > > ### Author Response · Authors · 2025-08-05
> > > **Thanks!**
> > >
> > > We thank the reviewer again for the great questions and for revising the rating! We will incorporate this rebuttal into the revision as the reviewer suggested.

---

### Official Review · Reviewer_C861 · 2025-07-04

**Clarity:** 3
**Significance:** 3
**Originality:** 3
**Rating:** 4
**Confidence:** 3

**Summary:**

The paper investigates online learning-based pricing policies in two-sided queueing markets with heterogeneous, price-sensitive arrivals, where customers and servers form matching pairs and depart from separate queues. The objective is to maximize platform profit while maintaining reasonable queue lengths, even when underlying demand and supply curves are unknown. The main contribution is the design and analysis of a dynamic, probabilistic pricing and matching policy that achieves a near-optimal tradeoff among regret $\tilde{O}(T^{1-\gamma})$, average queue length $\tilde{O}(T^{\gamma/2})$, and maximum queue length $\tilde{O}(T^\gamma)$. The paper provides theoretical analysis demonstrating improvements over prior learning-based approaches, and presents simulation results comparing the new method to relevant baselines.

**Questions:**

1. How does the method perform compared to recent RL/adaptive optimization baselines for online queue control? Why were these not included in the main experimental benchmark?

2. Is it possible to estimate the demand function using nonparametric regression and make decisions based on the resulting estimated steady-state matching rates?

**Ethical Concerns:**

["NO or VERY MINOR ethics concerns only"]

**Final Justification:**

Having reviewed the rebuttal, I believe the paper is suitable for acceptance at NeurIPS.

**Limitations:**

Yes

**Quality:**

3

**Strengths And Weaknesses:**

Strengths:

1. The model is well-motivated, addresses practical scenarios in online marketplaces, and is formalized with clear notation and reasonable economic and queueing assumptions. Assumptions are stated and discussed.

2. The proposed algorithm introduces a randomized two-price policy to resolve the tension between learning and queue management, resulting in provably superior performance metrics. This dynamic and probabilistic treatment of pricing is argued to be widely applicable.

3. The work provides a thorough theoretical analysis of the regret/queue length trade-off in learning-based pricing for two-sided queues. Regret and queue length bounds are carefully derived and supported by matching lower bounds, offering evidence for near-optimality up to log factors and strengthening the results in prior literature.

Weaknesses:

1. The proposed trade-off holds only for $\gamma \in (0, 1/6]$. For larger $\gamma$, the bound on regret does not improve (remains $\tilde{O}(T^{5/6})$), leaving a gap. The trade-off curve thus does not offer a smooth interpolation to the regime where queue lengths can be significantly higher for improved regret. Also, the regret bound hides large constants.

2. The multi-stage learning algorithm (incorporating projected stochastic gradient, repeated bisection search, and a complex exploration mechanism) involves numerous tunable parameters ($\eta$, $\delta$, $\epsilon$, $\alpha$, $q^\text{th}$) and repeated sample-collection via resets or altering queues. While the theoretical roles are justified, there is little discussion or analysis of runtime, computational costs, or scaling to large-scale practical systems (e.g., with many customer/server types).

---

> ### Author Rebuttal · Authors · 2025-07-30
>
> We thank the reviewer for the positive comments and valuable suggestions.
>
> **Comment:** "The multi-stage learning algorithm (incorporating projected stochastic gradient, repeated bisection search, and a complex exploration mechanism) involves numerous tunable parameters ($\eta$, $\delta$, $\epsilon$, $\alpha$, $q^{\mathrm{th}}$) and repeated sample-collection via resets or altering queues. While the theoretical roles are justified, there is little discussion or analysis of runtime, computational costs, or scaling to large-scale practical systems (e.g., with many customer/server types)."
>
> **Response:** The numerical results in Appendix M.1.4 (Figures 6,7,8) show that the tunable parameters can take a wide range of values while maintaining similar performance. Therefore, we can just set a $\gamma\in(0,1/6]$ according to our main results in Section 3 to balance the regret and queue lengths. Then the values of $\eta$, $\delta$, $\epsilon$, $\alpha$, and $q^{\mathrm{th}}$ will be determined by $\gamma$, as shown in Corollary 1 in Appendix A.2.
>
>
> The proposed pricing algorithm has computational complexity $\tilde{O}((I+J)T + IJT^{1-4\gamma})$, along with $T^{1-4\gamma}$ calls to a projection oracle, where $T$ is the time horizon, $I$ is the number of customer types, $J$ is the number of server types. The projection can be implemented by solving a convex quadratic program using interior point methods, which have a computational complexity of $\tilde{O}((IJ)^{3.5})$ [A]. Hence, the overall computational complexity is $\tilde{O}((I+J)T + (IJ)^{3.5} T^{1-4\gamma})$.
>
> The regret upper bound scales with the number of customer types $I$ and server types $J$ as $O(I^4J^4(I+J)T^{1-\gamma})$.
> The average queue length upper bound scales with $I$ and $J$ as $O(IJ(I+J)T^{\gamma/2})$.
> The maximum queue length does not scale with $I$ or $J$. We believe the dependencies on $I$ and $J$ can be improved, as our analysis mainly focuses on obtaining the correct dependence in terms of the time horizon $T$.
>
> We will add the above computational complexity and the scaling to the revision.
>
> **Comment:** "How does the method perform compared to recent RL/adaptive optimization baselines for online queue control? Why were these not included in the main experimental benchmark?"
>
> **Response:** Most works on reinforcement learning (RL) for queueing in the literature [B] [C] [D] [E] [F] do not utilize the special structure of our pricing and matching problem, and it is difficult to trade off between profit and queue length. When facing the curse of dimensionality of queueing systems, the works [B] [C] [E] use deep RL, which does not have any theoretical guarantee on the performance. On the other hand, our approach circumvents the curse of dimensionality by restricting to the set of probabilistic two-price policies as opposed to a fully dynamic policy, as we know that a two-price policy is near-optimal. This way, we only need to learn $O(I+J)$ parameters as opposed to $O(e^{I+J})$ for a fully dynamic policy.
>
>
> Other adaptive optimization approaches:
> Lipschitz bandit algorithms, such as those that discretize the price space and apply UCB-based methods, fail to balance matching rates and arrival rates on both sides, making it difficult to control queue lengths. Bandit convex optimization is also inapplicable, as the objective is nonconcave in prices, and the constraint set for balancing arrival rates is both nonconvex and unknown in terms of prices.
>
> We will add a discussion in the revision.
>
>
> **Comment:** "Is it possible to estimate the demand function using nonparametric regression and make decisions based on the resulting estimated steady-state matching rates?"
>
> **Response:** It is difficult because we are considering an online learning setting where learning and decision making occur simultaneously. Nonparametric regression does not address how to select a data collection policy.
> If we estimate the demand function by trying different prices without considering the arrival-rate balance or regret, both queue lengths and regret can grow linearly over time in the estimation process.
> For example, suppose that we try prices uniformly in the price space. For each queue, to obtain an accuracy of $\zeta$ for the estimation of the demand/supply function, we need to discretize the space into $\Theta(1/\zeta)$ points, and for each point, we need at least $\Theta(1/\zeta^2)$ samples according to the central limit theorem. In this process, the queue length will increase linearly over time, up to $\Theta(1/\zeta^3)$. Using the estimated functions, we can obtain a solution with accuracy $\zeta$, and we can use this solution until the time horizon $T$. Hence, the worst-case average queue length will be $\Theta(1/\zeta^3 + (T - 1/\zeta^3)\zeta)$. The lowest worst-case average queue length that this approach can achieve is $\Theta(T^{3/4})$ with $\zeta=T^{-1/4}$, while our approach can achieve a much smaller average queue length $\tilde{\Theta}(T^{1/12})$ with $\gamma=1/6$.
>
> There is a short discussion on Page 5, Lines 161-164, and we will add a more detailed discussion in the revision.
>
> **References**
>
> [A] Yurii Nesterov and Arkadii Nemirovskii. Interior-point polynomial algorithms in convex programming. SIAM, 1994.
>
> [B] Majid Raeis, Ali Tizghadam, and Alberto Leon-Garcia. Queue-learning: A reinforcement learning approach for providing quality of service. In Proceedings of the AAAI Conference on Artificial Intelligence, volume 35, pages 461–468, 2021.
>
> [C] Jim G Dai and Mark Gluzman. Queueing network controls via deep reinforcement learning. Stochastic Systems, 12(1):30–67, 2022.
>
> [D] Bai Liu, Qiaomin Xie, and Eytan Modiano. Rl-qn: A reinforcement learning framework for optimal control of queueing systems. ACM Transactions on Modeling and Performance Evaluation of Computing Systems, 7(1):1–35, 2022.
>
> [E] Haozhe Chen, Ang Li, Ethan Che, Jing Dong, Tianyi Peng, and Hongseok Namkoong. Qgym: Scalable simulation and benchmarking of queuing network controllers. Advances in Neural Information Processing Systems, 37:92401–92419, 2024.
>
> [F] Yashaswini Murthy, Isaac Grosof, Siva Theja Maguluri, and R. Srikant. Performance of npg in countable state-space average-cost rl, 2024.

---

> > ### Comment · Reviewer_C861 · 2025-08-06
> > **Reply to Authors**
> >
> > I appreciate the authors’ responses and the efforts made in the rebuttal. I will maintain my original score.

---

> > > ### Author Response · Authors · 2025-08-06
> > > **Thanks!**
> > >
> > > Thanks again for your positive comments! We hope our rebuttal adequately addresses your questions. Please don't hesitate to let us know if you have further questions/suggestions.

---

### Decision · Program_Chairs · 2025-09-17

**Decision:**

Accept (poster)

**Comment:**

**Summary**

The paper studies a two-sided market with price-sensitive customers and servers, aiming to maximize platform profit while controlling queues. It proposes an online-learning pricing policy that achieves near-optimal performance, improving regret–queue length trade-offs over prior work. The policy balances learning efficiency with system stability through dynamic and probabilistic mechanisms.

**Strengths**

The model studied in the paper is well motivated and the paper provides interesting results.

**Weaknesses**

There are only some minor issues raised by the Reviewers (see the reviews).

**Decision**

All the Reviewers agree that this is a good paper. Thus, I recommend **acceptance**.